# SemiNLL: A Framework of Noisy-Label Learning by Semi-Supervised Learning

**Zhuowei Wang**[1]    **Jing Jiang**[1]    **Bo Han**[2]    **Lei Feng**[3*]
**Bo An**[4]    **Gang Niu**[5]    **Guodong Long**[1]

[1]*University of Technology Sydney*   [2]*Hong Kong Baptist University*   [3]*Chongqing University*
[4]*Nanyang Technological University*   [5]*RIKEN Center for Advanced Intelligence Project*
*zhuowei.wang@student.uts.edu.au,   jing.jiang@uts.edu.au,   bhanml@comp.hkbu.edu.hk,   lfeng@cqu.edu.cn,*
*boan@ntu.edu.sg,   gang.niu.ml@gmail.com,   guodong.long@uts.edu.au*

**Reviewed on OpenReview:** *https: // openreview. net/ pdf? id= LTyqvLEv5b*

## Abstract

Deep learning with noisy labels is a challenging task, which has received much attention from the machine learning and computer vision communities. Recent prominent methods that build on a specific sample selection (SS) strategy and a specific semi-supervised learning (SSL) model achieved state-of-the-art performance. Intuitively, better performance could be achieved if stronger SS strategies and SSL models are employed. Following this intuition, one might easily derive various effective noisy-label learning methods using different combinations of SS strategies and SSL models, which is, however, simply reinventing the wheel in essence. To prevent this problem, we propose *SemiNLL*, a versatile framework that investigates how to naturally combine different SS and SSL components based on their effects and efficiencies. We conduct a systematic and detailed analysis of the combinations of possible components based on our framework. Our framework can absorb various SS strategies and SSL backbones, utilizing their power to achieve promising performance. The instantiations of our framework demonstrate substantial improvements over state-of-the-art methods on benchmark-simulated and real-world datasets with noisy labels.

## 1 Introduction

Deep Neural Networks (DNNs) have achieved great success in various real-world applications, such as image classification (Krizhevsky et al., 2012), detection (Ren et al., 2015), semantic segmentation (Long et al., 2015), anomaly detection (Liu et al., 2021), and reinforcement learning (Yu et al., 2020; Yang et al., 2021; 2022). Such a great success is demanding for large datasets with clean human-annotated labels. However, it is costly and time-consuming to correctly label massive images for building a large-scale dataset like ImageNet (Deng et al., 2009). Some common and less expensive ways to collect large datasets are through online search engines (Schroff et al., 2010) or crowdsourcing (Yu et al., 2018), which would, unfortunately, bring wrongly annotated labels to the collected datasets. Besides, an in-depth study (Zhang et al., 2016) showed that deep learning with noisy labels can lead to severe performance deterioration. Thus, it is crucial to alleviate the negative effects caused by noisy labels for training DNNs.

A typical strategy is to conduct *sample selection* (SS) and to train DNNs with selected samples (Han et al., 2018; Jiang et al., 2018; Song et al., 2019; Yu et al., 2019; Wei et al., 2020; Yao et al., 2020a; Xia et al., 2022). Since DNNs tend to learn simple patterns first before fitting noisy samples (Arpit et al., 2017), many studies utilize the small-loss trick, where the samples with smaller losses are taken as clean ones. For example, *Co-teaching* (Han et al., 2018) leverages two networks to select small-loss samples within each mini-batch for training each other. Later, Yu et al. (2019) pointed out the importance of the disagreement

---

*Corresponding author: Lei Feng.

between two networks and proposed *Co-teaching+*, which updates the two networks using the data on which the two networks hold different predictions. By contrast, *JoCoR* (Wei et al., 2020) proposes to reduce the diversity between two networks by training them simultaneously with a joint loss calculated from the selected small-loss samples. Although these methods have achieved satisfactory performance by training with selected small-loss samples, they simply discard other large-loss samples which may contain potentially useful information for the training process.

To make full use of all given samples, a prominent strategy is to consider selected samples as labeled "clean" data and other samples as unlabeled data, and to perform *semi-supervised learning* (SSL) (Laine & Aila, 2016; Tarvainen & Valpola, 2017; Berthelot et al., 2019; Arazo et al., 2020; Xie et al., 2020; Sohn et al., 2020; Zhang et al., 2021; Liu et al., 2022). Following this strategy, *SELF* (Nguyen et al., 2020) detects clean samples by removing noisy samples whose self-ensemble predictions of the model do not match the given labels in each iteration. With the selected labeled and unlabeled data, the problem becomes an SSL problem, and a *Mean-Teacher* model (Tarvainen & Valpola, 2017) can be trained. Another recent method, *DivideMix* (Li et al., 2020b), leverages Gaussian Mixture Model (GMM) (Permuter et al., 2006) to distinguish clean and noisy data. After removing the labels of noisy samples, *DivideMix* uses a strong SSL backbone called *MixMatch* (Berthelot et al., 2019) for training.

As shown above, both methods rely on a specific SS strategy and a specific SSL model. The two components play a vitally important role for combating label noise, and stronger components are expected to achieve better performance. This motivates us to investigate a versatile algorithmic framework that can leverage various SS strategies and SSL models. In this paper, we propose *SemiNLL*, which is a versatile framework to bridge the gap between SSL and *noisy-label learning* (NLL). Our framework can absorb various SS strategies and SSL backbones, utilizing their power to achieve promising performance. Guided by our framework, one can easily instantiate a specific learning algorithm for NLL, by specifying a commonly used SSL backbone with an SS strategy. The key contributions of our paper can be summarized as follows: (i) Our framework can not only provide an important prototype in the NLL community for further exploration into SS and SSL, but can also act as a conclusive work to prevent future researchers from simply reinventing the wheel. (ii) To instantiate our framework, we propose *DivideMix+* by replacing the epoch-level selection strategy of *DivideMix* (Li et al., 2020b) with a mini-batch level one. We also propose *GPL*, another instantiation of our framework that leverages a two-component $\boldsymbol{G}$*aussian mixture model* (Li et al., 2020b; Permuter et al., 2006) to select labeled (unlabeled) data and uses $\boldsymbol{P}$*seudo-*$\boldsymbol{L}$*abeling* (Arazo et al., 2020) as the SSL backbone. (iii) We conduct extensive experiments on benchmark-simulated and real-world datasets with noisy labels. Our instantiations, *DivideMix+* and *GPL*, outperform other state-of-the-art noisy-label learning methods. We also analyze the effects and efficiencies of different instantiations of our framework.

The rest of this paper is organized as follows. In Section 2, we first review the related works. Then, the overview of the framework is introduced in Section 3. Section 4 illustrates the instantiations of our framework in detail. After that, we demonstrate the experimental results in Section 5 and give detailed ablation studies and discussions in Section 6. The conclusion is in Section 7.

## 2 RELATED WORK

In this section, we briefly review several related aspects on which our framework builds.

### 2.1 Learning with noisy labels

For NLL, most of the existing methods could be roughly categorized into the following groups:

**Sample selection.** This family of methods regards samples with small loss as "clean" and trains the model only on selected clean samples. For example, *self-paced MentorNet* (Jiang et al., 2018), or equivalently *self-teaching*, selects small-loss samples and uses them to train the network by itself. To alleviate the sample-selection bias in *self-teaching*, Han et al. (2018) proposed an algorithm called *Co-teaching*, where two networks choose the next batch of data for each other for training based on the samples with smaller loss values. *Co-teaching+* (Yu et al., 2019) bridges the *disagreement strategy* (Malach & Shalev-Shwartz, 2017) with *Co-teaching* (Han et al., 2018) by updating the networks over data where two networks make different

predictions. In contrast, Wei et al. (2020) leveraged the agreement maximization algorithm (Kumar et al., 2010) by designing a joint loss to train two networks on the same mini-batch data and selected small-loss samples to update the parameters of both networks. The mini-batch SS strategy in our framework belongs to this direction. However, instead of ignoring the large-loss unclean samples, we just discard their labels and exploit the associated images in an SSL setup.

**Noise transition estimation.** Another line of NLL is to estimate the noise transition matrix for loss correction (Natarajan et al., 2013; Menon et al., 2015; Xiao et al., 2015; Goldberger & Ben-Reuven, 2016; Patrini et al., 2017; Hendrycks et al., 2018; Wang et al., 2020; Yao et al., 2020b; Wu et al., 2021a). Patrini et al. (2017) first estimated the noise transition matrix and trained the network with two different loss corrections. Hendrycks et al. (2018) proposed a loss correction technique that utilizes a small portion of trusted samples to estimate the noise transition matrix. Wang et al. (2020) proposed a model-agnostic approach to learn the transition matrix directly from data via meta-learning. However, the limitation of these methods is that they do not perform well on datasets with a large number of classes.

**Other deep learning methods.** Some other interesting and promising directions for NLL include meta-learning (Finn et al., 2017; Snell et al., 2017; Li et al., 2020a) based, pseudo-label estimation (Lee, 2013) based, and robust loss (Feng et al., 2020; Ghosh et al., 2017; Ma et al., 2020; Wang et al., 2019; Xu et al., 2019; Zhang & Sabuncu, 2018; Hu et al., 2021) based approaches. For meta-learning based approaches, most studies fall into two main categories: training a model that *adapts fast to different learning tasks* without overfitting to corrupted labels (Garcia et al., 2016; Li et al., 2019), and *learning to reweight* loss of each mini-batch to alleviate the adverse effects of corrupted labels (Ren et al., 2018; Shu et al., 2019; Zhang et al., 2020; Wu et al., 2021b; Zheng et al., 2021). Pseudo-label estimation based approaches reassign the labels for noisy samples. For example, *Joint-Optim* (Tanaka et al., 2018) corrects labels during training and updates network parameters simultaneously. The family of pseudo-label estimation has a close relationship with semi-supervised learning (Han et al., 2019; Lee, 2013; Tanaka et al., 2018; Yi & Wu, 2019). Robust loss based approaches focus on designing loss functions that are robust to noisy labels. Recently, some works (Wang et al., 2022; Xu et al., 2022) non-trivially extend NLL to federated setting (McMahan et al., 2017; Jiang et al., 2020), which is a distributed environment with non-IID data (Ma et al., 2022; Chen et al., 2022), privacy-preserving (Long et al., 2020; 2022), and heterogeneous model architecture (Tan et al., 2021) across clients or devices.

## 2.2 Semi-supervised learning

SSL methods leverage unlabeled data to provide additional information for the training model. A line of work is based on the concept of consistency regularization: if a perturbation is given to an unlabeled sample, the model predictions of the same sample should not be too different. Laine & Aila (2016) applied consistency between the output of the current network and the exponential moving average (EMA) of the output from the past epochs. Instead of averaging the model outputs, Tarvainen & Valpola (2017) proposed to update the network on every mini-batch using an EMA of model parameter values. Berthelot et al. (2019) introduced a holistic approach that well combines *MixUp* (Zhang et al., 2018), entropy minimization, and consistency regularization. Another line of SSL is pseudo-labeling, the objective of which is to generate pseudo-labels for unlabeled samples to enhance the learning process. Arazo et al. (2020) proposed a method to improve previous pseudo-labeling methods (Iscen et al., 2019) by adding *MixUp* augmentation (Zhang et al., 2018). Xie et al. (2020) and Sohn et al. (2020)used a confidence-based strategy pseudo labeling to select unlabeled data with high confidence. Zhang et al. (2021) proposed a curriculum learning method to leverage unlabeled data for SSL.

## 2.3 Combination of SS and SSL

Some previous studies that combine a specific SS strategy and a specific SSL backbone could be regarded as special cases in our framework. Ding et al. (2018) used a pre-trained DNN on the noisy dataset to select labeled samples. In the SSL stage, *Temporal Ensembling* (Laine & Aila, 2016) was used to handle labeled and unlabeled data. Nguyen et al. (2020) proposed a progressive noise filtering mechanism based on the *Mean-Teacher* model (Tarvainen & Valpola, 2017) and its self-ensemble prediction. Li et al. (2020b) used a

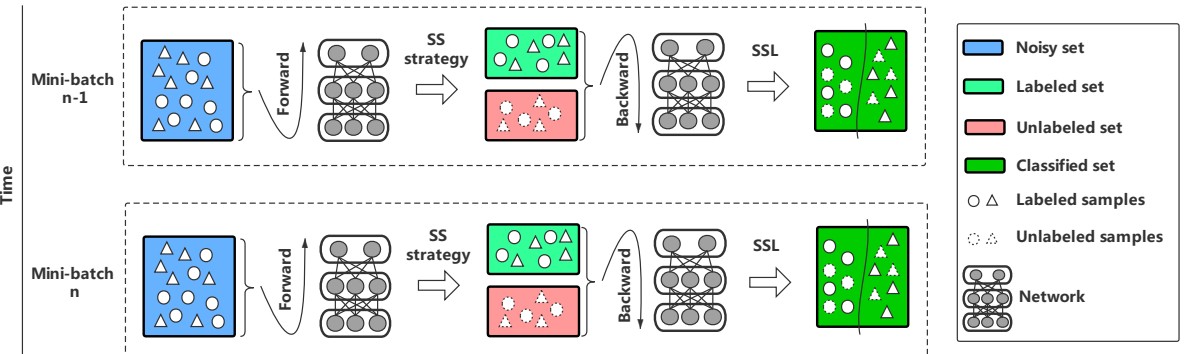

Figure 1: The schematic of *SemiNLL*. First, each mini-batch of data is forwarded to the network to conduct SS, which divides the original data into the labeled/unlabeled sets. Second, labeled/unlabeled samples are used to train the SSL backbone to produce accurate model output.

Gaussian Mixture Model (GMM) to divide noisy and clean samples based on their training losses and fitted them into a recent SSL algorithm called *MixMatch* (Berthelot et al., 2019). Each specific component used in these methods has its own pros and cons. This motivates us to propose a versatile framework that can build on various SS strategies and SSL backbones. In other words, many recent publications (Arazo et al., 2019; Li et al., 2020b; Nguyen et al., 2020) or preprints (Cordeiro et al., 2021; Wei et al., 2021b) could be taken as special instantiations of our framework, which indicates that a conclusive work like this paper is vitally necessary to prevent future researchers from simply reinventing the wheel.

## 3 The Overview of SemiNLL

In this section, we present *SemiNLL*, a versatile framework of learning with noisy labels by SSL. The idea behind our framework is that we effectively take advantage of the whole training set by trusting the labels of undoubtedly correct samples and utilizing only the image content of potentially corrupted samples. Previous sample selection methods (Han et al., 2018; Jiang et al., 2018; Yu et al., 2019; Wei et al., 2020) train the network only with selected clean samples, and they discard all potentially corrupted samples to avoid the harmful memorization of DNNs caused by the noisy labels of these samples. In this way, the feature information contained in the associated images might be discarded without being

---

**Algorithm 1** SemiNLL

**Input:** Network $f_\theta$, SS strategy SELECT, SSL method SEMI, epoch $T_{\max}$, iteration $I_{\max}$;
1: **for** $t = 1,2,\ldots,T_{\max}$ **do**
2:    **Shuffle** training set $\mathcal{D}_{train}$;
3:    **for** $n = 1,\ldots,I_{\max}$ **do**
4:       **Fetch** mini-batch $D_n$ from $\mathcal{D}_{train}$;
5:       **Obtain** $\mathcal{X}_m,\mathcal{U}_m \leftarrow$ SELECT$(D_n, f_\theta)$;
6:       **Update** $f_\theta \leftarrow$ SEMI$(\mathcal{X}_m,\mathcal{U}_m,f_\theta)$;
7:    **end for**
8: **end for**
**Output:** $f_\theta$

---

exploited. Our framework, alternatively, makes use of those corrupted samples by ignoring their labels while keeping the associated image content, transforming the NLL problem into an SSL setup. The mechanism of SSL that leverages labeled data to guide the learning of unlabeled data naturally fits well in training the model with the clean and noisy samples divided by our SS strategy. We first discuss the advantages of the mini-batch SS strategy in our framework and then introduce several SSL backbones used in our framework. The schematic of our framework is shown in Figure 1.

### 3.1 Mini-batch sample selection

During the SS process, a hazard called confirmation bias (Tarvainen & Valpola, 2017) is worth noting. Since the model is trained using the selected clean (labeled) and noisy (unlabeled) samples, wrongly selected clean samples in this iteration may keep being considered clean ones in the next iteration due to the model overfitting to their labels. Most existing methods (Li et al., 2020b; Nguyen et al., 2020) divide the whole

training set into the clean/noisy set on an epoch level. Our mini-batch SS strategy divides each mini-batch of samples into the clean subset $\mathcal{X}_m$ and the noisy subset $\mathcal{U}_m$ (Line 5 in Algorithm 1) right before updating the network using SSL backbones. The advantages of our mini-batch SS over epoch-wise SS are: (i) **Avoiding confirmation bias**: In the case of epoch-wise SS, the divided clean/noisy sets are incorporated into the SSL phase and will not be updated till the next epoch. Thus, the confirmation bias induced from those wrongly divided samples will accumulate within the whole epoch. Our mini-batch SS strategy divides each mini-batch of samples into clean/noisy batches before updating the network using SSL backbones. In the next mini-batch, the updated network can better distinguish clean and noisy samples, alleviating the confirmation bias mini-batch by mini-batch. (ii) **Improving computational efficiency**: Since the time complexity of most SS methods is not linear, the number of operations increases dramatically as the input size increases. Table 6.4 compares the training time of DividMix+ (mini-batch-wise) and DivideMix (epoch-wise) on CIFAR-10, showing DivideMix+ is more computationally efficient than DivideMix in both the SS process and the whole training process. (iii) **Injecting stochasticity**: During the epoch-wise SS process, the model tends to select the confident samples that have been selected in previous epochs due to the model overfitting their labels. In this way, some confident but noisy samples will keep being selected by the model, resulting in performance degradation. The mini-batch-wise SS can inject stochasticity in training since each mini-batch of data is randomly sampled from the whole dataset, avoiding the model constantly selecting the same confident samples. (iv) **Up-to-date model for SS**: The mini-batch-wise usage of SS and SSL makes the data selection up-to-date. The model used to select the clean samples is updated using the SSL method at each mini-batch.

### 3.2 SSL backbones

The mechanism of SSL that uses labeled data to guide the learning of unlabeled data fits well when dealing with clean/noisy data in NLL. The difference lies in an extra procedure, as introduced in Subsection 3.1, that divides the whole dataset into clean and noisy data. After the SS process, clean samples are considered labeled data and keep their annotated labels. The others are considered noisy samples, and their labels are discarded to be treated as unlabeled ones in SSL backbones. *SemiNLL* can build on a variety of SSL algorithms without any modifications to form an end-to-end training scheme for NLL. Concretely, we consider the following representative SSL backbones ranging from weak to strong according to their performance in SSL tasks: (i) *Temporal Ensembling* (Laine & Aila, 2016). The model uses an exponential moving average (EMA) of label predictions from the past epochs as a target for the unsupervised loss. It enforces consistency of predictions by minimizing the difference between the current outputs and the EMA outputs. (ii) *MixMatch* (Berthelot et al., 2019). MixMatch is a holistic method that combines *MixUp* (Zhang et al., 2018), entropy minimization, consistency regularization, and other traditional regularization tricks. It guesses low-entropy labels for augmented unlabeled samples and mixes labeled and unlabeled data using *MixUp* (Zhang et al., 2018). (iii) *Pseudo-Labeling* (Arazo et al., 2020). This method learns from unlabeled data by combining soft *pseudo-label* generation (Tanaka et al., 2018) and *MixUp* augmentation (Zhang et al., 2018) to reduce confirmation bias in training. In the next section, we will instantiate our framework by applying specific SS strategies and SSL backbones to the SELECT and SEMI placeholders in Algorithm 1.

## 4 The Instantiations of SemiNLL

### 4.1 Instantiation 1: DivideMix+

In Algorithm 1, if we (i) specify the SELECT placeholder as a GMM (Permuter et al., 2006), (ii) specify the SEMI placeholder as *MixMatch* (Berthelot et al., 2019) mentioned in Subsection 3.2, and (iii) train two independent networks wherein each network selects clean/noisy samples in the SS phase and predicts labels in the SSL phase for the other network, then our framework is instantiated into a mini-batch version of *DivideMix* (Li et al., 2020b). Specifically, during the SS process, *DivideMix* (Li et al., 2020b) fits a two-component GMM to the loss of each sample and obtains the posterior probability of a sample being clean or noisy. Since DNNs tend to learn simple patterns first before fitting noisy samples (Arpit et al., 2017), clean/noisy samples have lower/higher loss during the early epochs of training. Therefore, we can determine whether a sample is more likely to be clean or noisy based on its loss value. Given a classification loss $\ell_i$

(e.g., cross entropy) of sample $(\mathbf{x}_i, \tilde{\mathbf{y}}_i)$ in the dataset, we calculate the probability density function (pdf) of a mixture model of $k$ (in our case $k = 2$) components on the loss $\ell_i$ using Expectation-Maximization technique:

$$p(k \mid \ell_i) = \frac{p(k)\, p(\ell_i \mid k)}{p(\ell_i)}, \tag{1}$$

where $k = 0\,(1)$ denotes the clean (noisy) set, and $p(\text{clean} \mid \ell_i)$ is the posterior of the smaller-mean (clean) component of the GMM. The clean set $\mathcal{X}_e$ and the noisy set $\mathcal{U}_e$ can be defined as

$$\mathcal{X}_e = \{(\mathbf{x}_i, \tilde{\mathbf{y}}_i) : p\,(\text{clean} \mid \ell_i) \geq \tau\}, \quad \mathcal{U}_e = \{(\mathbf{x}_i, \tilde{\mathbf{y}}_i) : p\,(\text{clean} \mid \ell_i) < \tau\}, \tag{2}$$

where $\tau$ is a threshold on $p(\text{clean} \mid \ell_i)$. During the SSL phase, the clean set $\mathcal{X}_e$ and the noisy set $\mathcal{U}_e$ are fit into an improved *MixMatch* (Berthelot et al., 2019) strategy with label co-refinement and co-guessing. As shown in Figure 4 (b) in Appendix, the SS strategy (GMM) of *DivideMix* (Li et al., 2020b) is conducted on an epoch level. Since $\mathcal{X}_e$ and $\mathcal{U}_e$ are updated only once per epoch, the confirmation bias induced from the wrongly divided samples will be accumulated within the whole epoch. However, our mini-batch version, which is called *DivideMix+* (Figure 4 (c) in Appendix), divides each mini-batch of data into a clean subset $\mathcal{X}_m$ and a noisy subset $\mathcal{U}_m$, and updates the networks using the SSL backbone right afterwards. In the next mini-batch, the updated networks could better distinguish clean and noisy samples.

## 4.2 Instantiation 2: GPL

Intuitively, the choice of stronger SS strategies and SSL models would achieve better performance based on our framework. Thus, we still choose GMM to distinguish clean and noisy samples due to its flexibility in the sharpness of distribution (Li et al., 2020b). As for the SSL backbone, we choose the strongest *Pseudo-Labeling* (Arazo et al., 2020) introduced in Subsection 3.2. We call this instantiation *GPL* (***G****MM* + ***P****seudo-****L****abeling*). To keep our instantiation simple, we do not train two networks in *GPL* as in *DivideMix* (Li et al., 2020b) and *DivideMix+*. To our understanding, training two networks simultaneously might provide significant performance improvements. We will give detailed discussions about the effectiveness and efficiency of two networks in Section 6.5.

## 4.3 Self-prediction divider

Inspired by *SELF* (Nguyen et al., 2020), we introduce the *self-prediction divider*, a simple yet effective SS strategy which leverages the information provided by the network's own prediction to distinguish clean and noisy samples. Based on the phenomenon that DNN's predictions tend to be consistent on clean samples and inconsistent on noisy samples in different training iterations, we select the correctly annotated samples via the consistency between the original label set and the model's own predictions. In each mini-batch of data, if the model predicts the class with the highest likelihood to be the annotated label, the sample $(\mathbf{x}_i, \tilde{\mathbf{y}}_i)$ is divided into the clean set $\mathcal{X}_m$. Otherwise, the sample is divided into the noisy set $\mathcal{U}_m$. To be more specific, the model prediction $\hat{\mathbf{y}}_i$ is defined as

$$\hat{\mathbf{y}}_i \triangleq \arg\max_{j \in \{1, 2, \cdots, C\}} f(\mathbf{x}_i; \theta)[j], \tag{3}$$

where $C$ is the number of class, $\theta$ is the model parameters, and $f(\mathbf{x}_i; \theta)$ is the softmax output of the model. The clean set $\mathcal{X}_m$ and the noisy set $\mathcal{U}_m$ can be defined as

$$\mathcal{X}_m = \{(\mathbf{x}_i, \tilde{\mathbf{y}}_i) : \hat{\mathbf{y}}_i = \tilde{\mathbf{y}}_i\}, \quad \mathcal{U}_m = \{(\mathbf{x}_i, \tilde{\mathbf{y}}_i) : \hat{\mathbf{y}}_i \neq \tilde{\mathbf{y}}_i\}. \tag{4}$$

Compared to previous small-loss SS methods (Han et al., 2018; Wei et al., 2020; Yu et al., 2019), which depend on a known noise ratio to control how many small-loss samples should be selected in each training iteration, *self-prediction divider* does not need any additional information to perform SS strategy where the clean subset and the noisy subset are determined by the network itself. Concretely, we instantiate three learning algorithms by combining our ***self-p****rediction **d**ivider (SPD)* with three SSL backbones introduced in Subsection 3.2 and denote them as *SPD-Temporal Ensembling*, *SPD-MixMatch*, and *SPD-Pseudo-Labeling*, respectively.

### 4.4 Effects of the two components

This section demonstrates the effects of SS strategies and SSL backbones in our framework. To prove that a more robust SS strategy can boost performance for our framework, we propose *DivideMix-* (Figure 4) (a) by replacing the GMM in *DivideMix* (Li et al., 2020b) with our *self-prediction divider* on an epoch level. Since *self-prediction divider* is supposed to be weaker than GMM, *DivideMix-* is expected to achieve lower performance than *DivideMix* (Li et al., 2020b). To prove the effectiveness of the SSL backbone, we remove it after the SS process and only update the model using the supervised loss calculated from the clean samples. We will give detailed discussions in Section 6.

## 5 Experiment

### 5.1 Experiment setup

**Datasets.** We compare our method with five state-of-the-art algorithms in PU learning: We thoroughly evaluate our proposed *DivideMix+* and *GPL* on six datasets, including MNIST (Lecun et al., 1998), FASHION-MNIST (Xiao et al., 2017), CIFAR-10, CIFAR-100 (Krizhevsky & Hinton, 2009), Clothing1M (Xiao et al., 2015), and WebVision (Li et al., 2017). The detailed characteristics of the datasets in the experiments are shown in Table 9. MNIST and FASHION-MNIST contain 60K training images and 10K test images of size $28 \times 28$. CIFAR-10 and CIFAR-100 contain 50K training images and 10K test images of size $32 \times 32$ with three channels. According to previous studies (Zhang & Sabuncu, 2018; Li et al., 2020b; Wei et al., 2020), we experiment with two types of label noise: symmetric noise and asymmetric noise. Symmetric label noise is produced by changing the original label to all possible labels randomly and uniformly according to the noise ratio. Asymmetric label noise is similar to real-world noise, where labels are flipped to similar classes. Clothing1M is a large-scale real-world dataset that consists of one million training images from online shopping websites with labels annotated from surrounding texts. The estimated noise ratio is about 40%. WebVision is a large-scale dataset which consists of real-world web noise. We follow Chen et al. (2019); Li et al. (2020b) to create a mini version of WebVision that uses the Google subset images of the top 50 classes.

**Network structure and optimizer.** Following previous works (Arazo et al., 2020; Li et al., 2020b; Wei et al., 2020; Zhang & Sabuncu, 2018), we use a 2-layer MLP for MNIST, a ResNet-18 (He et al., 2016) for FASHION-MNIST, the well-known "13-CNN" architecture (Tarvainen & Valpola, 2017) for CIFAR-10 and CIFAR-100, a ResNet-50 (He et al., 2016) for Clothing1M and the inception-resnet v2 (Szegedy et al., 2017) for mini WebVision. To ensure a fair comparison between the instantiations of our framework and other methods, we keep the training settings for MNIST, CIFAR-10, CIFAR-100, Clothing1M, and mini WebVision as close as possible to *DivideMix* (Li et al., 2020b) and FASHION-MNIST close to *GCE* (Zhang & Sabuncu, 2018). For FASHION-MNIST, the network is trained using stochastic gradient descent (SGD) with 0.9 momentum and a weight decay of $1 \times 10^{-4}$ for 120 epochs. For MNIST, CIFAR-10, and CIFAR-100, all networks are trained using SGD with 0.9 momentum and a weight decay of $5 \times 10^{-4}$ for 300 epochs.

**Baselines.** We compare *DivideMix+* and *GPL* to previous state-of-the-art algorithms from *Co-teaching* (Han et al., 2018), *F-correction* (Patrini et al., 2017), *GCE* (Zhang & Sabuncu, 2018), *M-correction* (Arazo et al., 2019), and *DivideMix* (Li et al., 2020b). We implement all methods by PyTorch on NVIDIA Tesla V100 GPUs. (1) *Cross-Entropy*, which trains the network using the cross-entropy loss. (2) *Coteaching* (Han et al., 2018), which trains two networks and cross-updates the parameters of peer networks. (3) *F-correction* (Patrini et al., 2017), which corrects the prediction by the label transition matrix. As suggested by the authors, we first train a standard network using the cross-entropy loss to estimate the transition matrix. (4) *JoCoR* (Wei et al., 2020), which uses a joint loss to train two networks on the same mini-batch data and selects small-loss samples to update the networks. (5) *GCE* (Zhang & Sabuncu, 2018), which uses a theoretically grounded and easy-to-use loss function, the $\mathcal{L}_q$ loss, for NLL. (6) *M-correction* (Arazo et al., 2019), which models clean and noisy samples by fitting a two-component BMM and applies *MixUp* data augmentation (Zhang et al., 2018). (7) *DivideMix* (Li et al., 2020b), which divides clean and noisy samples by using a GMM on an epoch level and leverages *MixMatch* (Berthelot et al., 2019) as the SSL backbone.

Table 1: Average test accuracy (%) and standard deviation (5 runs) in various datasets under symmetric label noise. The best accuracy is **bold-faced**. The second-best accuracy is underlined.

| Datasets | Method | Symmetric | | | | Mean |
| | | Noise ratio | | | | |
| | | 20% | 40% | 60% | 80% | |
|---|---|---|---|---|---|---|
| MNIST | Cross-Entropy | 86.16 ± 0.34 | 70.39 ± 0.59 | 50.35 ± 0.51 | 23.41 ± 0.96 | 57.58 |
| | Co-teaching | 91.20 ± 0.03 | 90.02 ± 0.02 | 83.21 ± 0.71 | 25.33 ± 0.84 | 72.44 |
| | F-correction | 93.93 ± 0.10 | 84.30 ± 0.43 | 65.06 ± 0.64 | 29.81 ± 0.63 | 68.27 |
| | JoCoR | 94.30 ± 0.09 | 92.66 ± 0.13 | 89.94 ± 0.24 | 75.37 ± 0.74 | 88.07 |
| | GCE | 94.36 ± 0.11 | 93.61 ± 0.17 | 92.46 ± 0.20 | 85.04 ± 0.66 | 91.37 |
| | M-correction | **97.25 ± 0.03** | 96.63 ± 0.04 | 95.07 ± 0.08 | 86.19 ± 0.42 | 93.79 |
| | DivideMix | 96.80 ± 0.08 | 96.53 ± 0.06 | 96.47 ± 0.04 | 95.15 ± 0.25 | 96.24 |
| | **GPL (ours)** | 96.67 ± 0.09 | 96.27 ± 0.08 | 95.82 ± 0.09 | 94.81 ± 0.15 | 95.89 |
| | **DivideMix+ (ours)** | 96.83 ± 0.06 | **96.79 ± 0.06** | **96.69 ± 0.03** | **95.91 ± 0.10** | **96.56** |
| FASHION MNIST | Cross-Entropy | 90.83 ± 0.26 | 86.44 ± 0.11 | 77.27 ± 0.56 | 61.84 ± 1.27 | 79.10 |
| | Co-teaching | 89.18 ± 0.32 | 89.13 ± 0.05 | 80.08 ± 0.25 | 60.36 ± 2.15 | 79.69 |
| | F-correction | **93.37 ± 0.17** | 92.27 ± 0.06 | 90.32 ± 0.30 | 85.78 ± 0.06 | 90.43 |
| | JoCoR | 91.43 ± 0.14 | 90.55 ± 0.11 | 86.89 ± 0.29 | 79.61 ± 0.41 | 87.12 |
| | GCE | 93.35 ± 0.09 | 92.58 ± 0.11 | 91.30 ± 0.20 | 88.01 ± 0.22 | 91.31 |
| | M-correction | 93.03 ± 0.15 | 92.74 ± 0.42 | 91.61 ± 0.02 | 85.25 ± 0.23 | 90.66 |
| | DivideMix | 92.98 ± 0.17 | 92.55 ± 0.13 | 91.55 ± 0.31 | 88.55 ± 0.24 | 90.66 |
| | **GPL (ours)** | 92.94 ± 0.20 | 91.38 ± 0.54 | 89.97 ± 0.16 | 87.14 ± 0.65 | 90.36 |
| | **DivideMix+ (ours)** | 93.20 ± 0.08 | **92.89 ± 0.15** | **92.15 ± 0.16** | **88.70 ± 0.17** | **91.74** |
| CIFAR-10 | Cross-Entropy | 83.48 ± 0.17 | 68.49 ± 0.40 | 48.65 ± 0.06 | 27.56 ± 0.43 | 57.05 |
| | Co-teaching | 67.73 ± 0.71 | 62.83 ± 0.72 | 48.81 ± 0.78 | 27.56 ± 2.71 | 51.73 |
| | F-correction | 83.27 ± 0.04 | 73.67 ± 0.30 | 77.64 ± 0.11 | 63.95 ± 0.32 | 74.63 |
| | JoCoR | 85.22 ± 0.06 | 80.27 ± 0.37 | 58.72 ± 0.29 | 29.67 ± 0.68 | 63.47 |
| | GCE | 89.72 ± 0.10 | 87.75 ± 0.05 | 84.11 ± 0.26 | 72.84 ± 0.30 | 83.61 |
| | M-correction | 92.01 ± 0.40 | 90.09 ± 0.68 | 85.90 ± 0.22 | 70.57 ± 0.85 | 84.64 |
| | DivideMix | 94.82 ± 0.09 | 93.95 ± 0.14 | 92.28 ± 0.08 | 89.30 ± 0.17 | 92.59 |
| | **GPL (ours)** | 94.45 ± 0.20 | 94.00 ± 0.22 | **93.32 ± 0.10** | 91.76 ± 0.23 | 93.38 |
| | **DivideMix+ (ours)** | **94.84 ± 0.12** | **94.03 ± 0.20** | 93.08 ± 0.19 | **91.91 ± 0.07** | **93.47** |
| CIFAR-100 | Cross-Entropy | 60.93 ± 0.40 | 46.24 ± 0.74 | 29.00 ± 0.38 | 11.42 ± 0.19 | 36.90 |
| | F-correction | 60.49 ± 0.29 | 48.93 ± 0.21 | 48.74 ± 0.41 | 22.93 ± 0.78 | 45.27 |
| | JoCoR | 65.89 ± 0.08 | 49.65 ± 0.51 | 32.38 ± 0.60 | 16.91 ± 0.63 | 41.21 |
| | GCE | 69.20 ± 0.10 | 65.90 ± 0.25 | 57.33 ± 0.18 | 18.19 ± 1.15 | 52.66 |
| | M-correction | 67.96 ± 0.17 | 64.48 ± 0.76 | 55.37 ± 0.72 | 24.21 ± 1.06 | 53.01 |
| | DivideMix | 73.17 ± 0.28 | 71.01 ± 0.16 | 66.61 ± 0.18 | 43.25 ± 0.82 | 63.51 |
| | **GPL (ours)** | 71.24 ± 0.24 | 68.89 ± 0.07 | 65.80 ± 0.63 | **59.96 ± 0.15** | 66.47 |
| | **DivideMix+ (ours)** | **73.22 ± 0.21** | **71.03 ± 0.32** | **67.52 ± 0.19** | 58.07 ± 0.71 | **67.46** |

## 5.2 Performance comparison

The results of all the methods under symmetric and asymmetric noise types on MNIST, FASHION-MNIST, CIFAR-10, and CIFAR-100 are shown in Table 1 and Table 2. The results on Clothing1M and mini WebVision are shown in Table 3 and Table 4. Furthermore, We delve into the reasons beyond these results in Section 6.

**Results on MNIST.** *DivideMix+* surpasses *DivideMix* across symmetric and asymmetric noise at all noise ratios, showing the effectiveness of the mini-batch SS strategy in our framework. *GPL* performs worse than DivideMix under symmetric noise while outperforming it under symmetric noise. In the cases of Symmetric 20% and 40%, *DivideMix+* and *M-correction* perform better than the other methods. When it comes to Asymmetric noise, *GPL* and *M-correction* demonstrate better performance. However, the performance of *M-correction* drops dramatically in the more challenging Symmetric 80% case where *DivideMix+* surpasses all the other algorithms.

**Results on FASHION-MNIST.** FASHION-MNIST is quite similar to MNIST but more complicated. *DivideMix+* still outperforms *DivideMix* on symmetric and asymmetric noise at all noise ratios. *GPL* still performs worse than DivideMix under symmetric noise and better under symmetric noise. Under symmetric

Table 2: Average test accuracy (%) and standard deviation (5 runs) in various datasets under asymmetric label noise. The best accuracy is **bold-faced**. The second-best accuracy is underlined.

| Datasets | Method | Asymmetric | | | | Mean |
|---|---|---|---|---|---|---|
| | | Noise ratio | | | | |
| | | 10% | 20% | 30% | 40% | |
| MNIST | Cross-Entropy | $95.78 \pm 0.19$ | $91.15 \pm 0.26$ | $86.01 \pm 0.25$ | $79.92 \pm 0.32$ | 88.22 |
| | Co-teaching | $90.32 \pm 0.02$ | $89.03 \pm 0.02$ | $79.80 \pm 0.27$ | $64.94 \pm 0.02$ | 81.02 |
| | F-correction | $96.39 \pm 0.04$ | $94.27 \pm 0.21$ | $89.33 \pm 0.94$ | $81.61 \pm 0.42$ | 90.40 |
| | JoCoR | $95.43 \pm 0.04$ | $94.39 \pm 0.13$ | $90.15 \pm 0.24$ | $87.31 \pm 0.05$ | 91.82 |
| | GCE | $94.61 \pm 0.13$ | $94.43 \pm 0.07$ | $94.00 \pm 0.12$ | $93.42 \pm 0.12$ | 94.12 |
| | M-correction | $\underline{96.74 \pm 0.03}$ | $\underline{96.70 \pm 0.10}$ | $\mathbf{96.67 \pm 0.07}$ | $94.85 \pm 0.40$ | 96.24 |
| | DivideMix | $96.17 \pm 0.06$ | $96.11 \pm 0.09$ | $95.88 \pm 0.05$ | $95.83 \pm 0.05$ | 96.00 |
| | **GPL (ours)** | $\mathbf{96.76 \pm 0.04}$ | $\mathbf{96.71 \pm 0.03}$ | $96.49 \pm 0.08$ | $\underline{96.45 \pm 0.04}$ | **96.60** |
| | **DivideMix+ (ours)** | $96.67 \pm 0.04$ | $96.66 \pm 0.07$ | $\underline{96.50 \pm 0.04}$ | $\mathbf{96.46 \pm 0.04}$ | $\underline{96.57}$ |
| FASHION MNIST | Cross-Entropy | $\underline{93.88 \pm 0.16}$ | $92.20 \pm 0.33$ | $90.41 \pm 0.67$ | $84.56 \pm 0.41$ | 90.26 |
| | Co-teaching | $88.01 \pm 0.03$ | $78.88 \pm 0.20$ | $70.07 \pm 0.38$ | $61.97 \pm 0.21$ | 74.73 |
| | F-correction | $\mathbf{94.17 \pm 0.12}$ | $\mathbf{93.88 \pm 0.10}$ | $\mathbf{93.50 \pm 0.10}$ | $\mathbf{93.25 \pm 0.16}$ | **93.7** |
| | JoCoR | $91.54 \pm 0.13$ | $88.60 \pm 0.47$ | $84.37 \pm 0.24$ | $81.68 \pm 0.62$ | 86.55 |
| | GCE | $93.51 \pm 0.17$ | $\underline{93.24 \pm 0.14}$ | $\underline{92.21 \pm 0.27}$ | $89.53 \pm 0.53$ | 92.12 |
| | M-correction | $92.11 \pm 0.93$ | $91.26 \pm 1.35$ | $89.79 \pm 1.28$ | $89.58 \pm 2.20$ | 90.69 |
| | DivideMix | $91.83 \pm 0.24$ | $91.09 \pm 0.08$ | $89.90 \pm 0.26$ | $87.58 \pm 0.26$ | 90.10 |
| | **GPL (ours)** | $92.52 \pm 0.22$ | $92.23 \pm 0.09$ | $92.15 \pm 0.26$ | $\underline{91.64 \pm 0.31}$ | $\underline{92.14}$ |
| | **DivideMix+ (ours)** | $92.56 \pm 0.39$ | $92.25 \pm 0.21$ | $91.62 \pm 0.08$ | $89.67 \pm 0.44$ | 91.53 |
| CIFAR-10 | Cross-Entropy | $90.85 \pm 0.06$ | $87.23 \pm 0.40$ | $81.92 \pm 0.32$ | $76.23 \pm 0.45$ | 84.06 |
| | Co-teaching | $62.85 \pm 2.20$ | $61.04 \pm 1.31$ | $54.50 \pm 0.39$ | $51.68 \pm 1.66$ | 57.52 |
| | F-correction | $89.79 \pm 0.33$ | $86.79 \pm 0.67$ | $83.34 \pm 0.30$ | $76.81 \pm 1.08$ | 84.18 |
| | JoCoR | $88.62 \pm 0.21$ | $89.79 \pm 0.17$ | $82.37 \pm 0.12$ | $77.90 \pm 0.69$ | 84.67 |
| | GCE | $90.40 \pm 0.09$ | $89.30 \pm 0.13$ | $86.89 \pm 0.22$ | $82.60 \pm 0.17$ | 87.30 |
| | M-correction | $92.28 \pm 0.12$ | $92.13 \pm 0.17$ | $91.38 \pm 0.11$ | $90.43 \pm 0.23$ | 91.56 |
| | DivideMix | $93.61 \pm 0.15$ | $92.99 \pm 0.21$ | $91.79 \pm 0.36$ | $90.57 \pm 0.31$ | 92.24 |
| | **GPL (ours)** | $\mathbf{94.32 \pm 0.01}$ | $\mathbf{94.23 \pm 0.07}$ | $\mathbf{93.79 \pm 0.06}$ | $\mathbf{93.02 \pm 0.30}$ | **93.84** |
| | **DivideMix+ (ours)** | $\underline{94.27 \pm 0.23}$ | $\underline{93.92 \pm 0.20}$ | $\underline{92.82 \pm 0.28}$ | $\underline{91.91 \pm 0.24}$ | $\underline{93.23}$ |
| CIFAR-100 | Cross-Entropy | $68.58 \pm 0.34$ | $68.82 \pm 0.22$ | $53.99 \pm 0.50$ | $44.31 \pm 0.23$ | 58.93 |
| | F-correction | $68.87 \pm 0.06$ | $64.11 \pm 0.37$ | $56.45 \pm 0.59$ | $46.44 \pm 0.50$ | 58.97 |
| | JoCoR | $69.44 \pm 0.16$ | $66.91 \pm 0.54$ | $54.71 \pm 0.42$ | $39.76 \pm 0.97$ | 57.71 |
| | GCE | $70.77 \pm 0.14$ | $69.22 \pm 0.15$ | $64.60 \pm 0.25$ | $51.72 \pm 1.17$ | 64.08 |
| | M-correction | $69.44 \pm 0.52$ | $67.25 \pm 0.81$ | $63.16 \pm 1.55$ | $52.90 \pm 1.79$ | 63.19 |
| | DivideMix | $\mathbf{74.00 \pm 0.29}$ | $\underline{73.28 \pm 0.42}$ | $\mathbf{72.84 \pm 0.36}$ | $54.33 \pm 0.69$ | 68.61 |
| | **GPL (ours)** | $71.94 \pm 0.29$ | $71.22 \pm 0.11$ | $70.56 \pm 0.23$ | $\mathbf{69.84 \pm 0.41}$ | **70.89** |
| | **DivideMix+ (ours)** | $\underline{73.49 \pm 0.31}$ | $\mathbf{73.30 \pm 0.22}$ | $\underline{72.36 \pm 0.43}$ | $\underline{55.63 \pm 0.60}$ | $\underline{68.70}$ |

noise, *DivideMix+* outperforms most of other methods, while *F-correction* and *GCE* surprisingly achieve comparable test accuracy under asymmetric noise.

**Results on CIFAR-10.** *DivideMix+* constantly outperforms *DivideMix*, especially in the cases with higher noise ratios. *DivideMix+* achieves an improvement in the accuracy of +2.61% in Symmetric 80% and +1.34% in Asymmetric 40% over *DivideMix*. We believe the reason is that the mini-batch SS strategy used in our framework can better mitigate the confirmation bias induced from wrongly divided samples in more challenging scenarios. In the easiest Symmetric 20%, 40%, *DivideMix*, *DivideMix+* and *GPL* tie closely with *DivideMix+* slightly working

Table 3: Test accuracy (%) on Clothing1M.

| Methods | Test Accuracy |
|---|---|
| Cross-Entropy | 69.21 |
| F-correction (Patrini et al., 2017) | 69.84 |
| M-correction (Arazo et al., 2019) | 71.00 |
| Joint-Optim (Tanaka et al., 2018) | 72.16 |
| Meta-Learning (Li et al., 2019) | 73.47 |
| PENCIL (Yi & Wu, 2019) | 73.49 |
| Dividemix (Li et al., 2020b) | 73.91 |
| GPL(ours) | 73.89 |
| Dividemix+(ours) | **74.14** |

better than the other two. In the harder cases (symmetric 60%, symmetric 80% and all asymmetric cases), *GPL* and *DivideMix+* surpass the other methods over a large margin. *GPL* shows superior performance under asymmetric noise.

**Results on CIFAR-100.** There are 100 classes in CIFAR-100, making it more challenging to train than CIFAR-10. *Coteaching* tends to fail in CIFAR-100 even under low noise ratios. In most cases, *DivideMix+* and *DivideMix* achieve higher test accuracy than the other approaches, with *DivideMix+* performing better. Specifically, *DivideMix+* surpasses *DivideMix* by 14.82% in the hardest symmetric 80% case. *GPL* is sur-

passed by DivideMix in most cases except for the hardest symmetric 80% and asymmetric 40% cases. An interesting phenomenon is that all the approaches suffer from performance deterioration in the asymmetric 40% cases except *GPL*, which significantly outperforms the second-best algorithm over +14%.

**Results on real-world datasets.** To verify the robustness of our framework under real-world datasets with noisy labels, we demonstrate the effectiveness of *DivideMix+* and *GPL* on Clothing1M and (mini) WebVision. As shown in Table 3, the performance of *DivideMix+* is better than that of *DivideMix* and other methods. *GPL* is slightly lower than DivideMix without using two networks for training. In Table 4, both *DivideMix+* and *GPL* outperform compared methods.

Table 4: Test accuracy (%) on (mini) WebVision.

| Methods | Test Accuracy |
|---|---|
| F-correction (Patrini et al., 2017) | 61.12 |
| Decoupling (Malach & Shalev-Shwartz, 2017) | 62.54 |
| D2L (Ma et al., 2018) | 62.68 |
| MentorNet (Jiang et al., 2018) | 63.00 |
| Co-teaching (Han et al., 2018) | 63.58 |
| Iterative-CV (Chen et al., 2019) | 65.24 |
| Dividemix (Li et al., 2020b) | 77.32 |
| GPL(ours) | 77.84 |
| Dividemix+(ours) | **78.28** |

## 6 Ablation Studies and Discussions

In this section, we investigate our framework in depth to gain more insights. Specifically, we analyze the pros and cons of different SS and SSL components, and also analyze the effects and efficiencies of the instantiations of our framework, which sheds light on how SSL components affect SS process. Moreover, we conduct an ablation study on training with two networks.

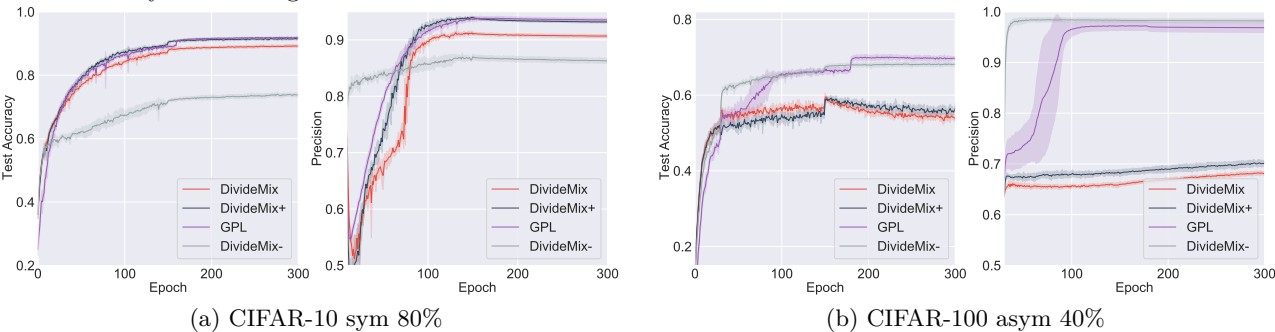

(a) CIFAR-10 sym 80%     (b) CIFAR-100 asym 40%

Figure 2: Results of ablation study. For (a) and (b), left: accuracy vs. epochs; right: precision vs. epochs.

### 6.1 The effects of mini-batch mechanism

In Table 5, the test accuracy of *DivideMix+* constantly outperforms *DivideMix* in most cases. This is because our mini-batch mechanism injects more stochasticity in training and reduces the harm of confirmation bias. In Figure 2 (a) and (b), the precision of *DivideMix+* outperforms *DivideMix* during the training process, showing that our approach is better at finding clean instances. We also compare GPL (mini-batch) with GPL (epoch) on CIFAR-10 in Appendix D.4.

### 6.2 The effects of different SS strategies

To study how SS strategies can affect the performance of our framework, we propose *DivideMix-* by replacing the GMM component in *DivideMix* with our *self-prediction divider* yet maintaining the epoch-level SS strategy for a fair comparison. In CIFAR-10, the difference between *DivideMix-* and *DivideMix* is not obvious in the lower noise ratios. However, in the most difficult symmetric 80% case, the test accuracy of *DivideMix* is +12.69% higher than *DivideMix-* and the precision of *DivideMix* in Figure 2 (a) outperforms *DivideMix-*. In CIFAR-100, the difference of test accuracy is even greater under symmetric noise. An impressive phenomenon to note is that *DivideMix-* excels in the asymmetric 40% case in CIFAR-100 with the highest precision and accuracy. This is because *GMM* distinguishes clean and noisy samples by fitting their loss distribution, which works effectively for symmetric noise. However, for asymmetric noise, most samples have near-zero normalized loss due to the low entropy predictions of the network, causing performance

Table 5: Test accuracy (%) of *DivideMix-*, *DivideMix*, and *DivideMix+*.

| Datasets | Method | Symmetric | | | | Asymmetric | | | |
|---|---|---|---|---|---|---|---|---|---|
| | | Noise ratio | | | | Noise ratio | | | |
| | | 20% | 40% | 60% | 80% | 10% | 20% | 30% | 40% |
| CIFAR-10 | DivideMix- | $94.49 \pm 0.02$ | $93.64 \pm 0.12$ | $91.65 \pm 0.34$ | $76.61 \pm 1.26$ | $93.58 \pm 0.02$ | $92.87 \pm 0.14$ | $91.21 \pm 0.21$ | $90.42 \pm 0.23$ |
| | DivideMix | $94.82 \pm 0.09$ | $93.95 \pm 0.14$ | $92.28 \pm 0.08$ | $89.30 \pm 0.17$ | $93.61 \pm 0.15$ | $92.99 \pm 0.21$ | $91.79 \pm 0.36$ | $90.57 \pm 0.31$ |
| | DivideMix+ (ours) | $\mathbf{94.84 \pm 0.12}$ | $\mathbf{94.03 \pm 0.20}$ | $\mathbf{93.08 \pm 0.19}$ | $\mathbf{91.91 \pm 0.07}$ | $\mathbf{94.27 \pm 0.23}$ | $\mathbf{93.92 \pm 0.20}$ | $\mathbf{92.82 \pm 0.28}$ | $\mathbf{91.91 \pm 0.24}$ |
| CIFAR-100 | DivideMix- | $72.51 \pm 0.32$ | $69.27 \pm 0.46$ | $61.13 \pm 0.60$ | $25.96 \pm 0.78$ | $73.62 \pm 0.12$ | $72.32 \pm 0.24$ | $70.64 \pm 0.20$ | $\mathbf{68.04 \pm 1.24}$ |
| | DivideMix | $73.17 \pm 0.28$ | $71.01 \pm 0.16$ | $66.61 \pm 0.18$ | $43.25 \pm 0.82$ | $\mathbf{74.00 \pm 0.29}$ | $73.28 \pm 0.42$ | $\mathbf{72.84 \pm 0.36}$ | $54.33 \pm 0.69$ |
| | DivideMix+ (ours) | $\mathbf{73.22 \pm 0.21}$ | $\mathbf{71.03 \pm 0.32}$ | $\mathbf{67.52 \pm 0.19}$ | $\mathbf{58.07 \pm 0.71}$ | $73.49 \pm 0.31$ | $\mathbf{73.30 \pm 0.22}$ | $72.36 \pm 0.43$ | $55.63 \pm 0.60$ |

Table 6: Test accuracy (%) of the baseline and three SSL backbones integrated into our proposed framework.

| Dataset | CIFAR-10 | | | CIFAR-100 | | |
|---|---|---|---|---|---|---|
| Method/Noise rate | 20% | 50% | 80% | 20% | 50% | 80% |
| SPD-Cross-Entropy | $83.13 \pm 0.16$ | $79.74 \pm 0.10$ | $49.14 \pm 0.15$ | $45.07 \pm 0.55$ | $35.02 \pm 0.57$ | $10.22 \pm 0.10$ |
| SPD-Temporal Ensembling | $83.15 \pm 0.06$ | $80.16 \pm 0.36$ | $49.10 \pm 0.13$ | $46.16 \pm 0.12$ | $39.91 \pm 0.60$ | $12.37 \pm 0.67$ |
| SPD-MixMatch | $93.53 \pm 0.52$ | $90.22 \pm 0.18$ | $88.77 \pm 0.20$ | $72.89 \pm 0.30$ | $68.57 \pm 0.20$ | $33.92 \pm 0.20$ |
| SPD-Pseudo-Labeling | $\mathbf{94.52 \pm 0.06}$ | $\mathbf{93.24 \pm 0.36}$ | $\mathbf{90.27 \pm 0.34}$ | $\mathbf{73.84 \pm 0.48}$ | $\mathbf{68.61 \pm 0.40}$ | $\mathbf{55.37 \pm 0.34}$ |

deterioration for *GMM*. Since *SPD* leverages the prediction of its own network to choose clean samples, its performance is more sensitive to noisy rates rather than noise type.

## 6.3 The effects of different SSL backbones

We evaluate the effects of SSL backbones in our framework by combining the *self-prediction divider (SPD)* with three different SSL methods and a baseline which only updates the model using the cross-entropy loss calculated from clean samples. We denote them as *SPD-Temporal Ensembling (TE)*, *SPD-MixMatch (MM)*, *SPD-Pseudo-Labeling (PL)*, and *SPD-Cross-Entropy (CE)*, respectively. For fair comparisons, we use the "13-CNN" architecture (Tarvainen & Valpola, 2017) for all methods across different datasets. We keep most hyperparameters introduced by the SSL methods close to their original papers (Arazo et al., 2020; Berthelot et al., 2019; Laine & Aila, 2016), since they can be easily integrated into our framework without major adjustments. In Table 6, *SPD-MM* and *SPD-PL* outperform *SPD-TE* by a large domain in both CIFAR-10 and CIFAR-100, especially under 80% noise ratio. This is reasonable because *TE* (Laine & Aila, 2016) only uses consistency regularization for unsupervised loss, while *MM* (Berthelot et al., 2019) and *PL* (Arazo et al., 2020) also leverage entropy regularization as well as *MixUp* data augmentation (Zhang et al., 2018). Moreover, *SPD-PL* achieves remarkable test accuracy under 80% noise ratio in CIFAR-100. This is due to the additional loss used in *SPD-PL* that prevents the model from assigning all labels to a single class at the early training stage. From the results of *SPD-CE*, we can see that after the removal of the SSL backbone, the test accuracy drops dramatically compared to *SPD-MM* and *SPD-PL*. This is due to the substantial amount of data that has been removed by the *SPD*, leaving very few samples per class. Thus, instead of discarding noisy samples, using them as unlabeled ones in SSL backbones is an effective way to combat noisy labels.

The benefits of using SSL methods in our framework are in these aspects: (i) improving SS performance incrementally and (ii) leveraging the potentially noisy samples to learn feature representation. **For (i)**, as shown in Figure 2 (a) and (b), the precision of *GPL* is higher than *DivideMix+* and *DivideMix*. This is because the SSL backbone used in *GPL*, i.e., *Pseudo-Labeling*, can better alleviate the confirmation bias induced during the SS process and improve the prediction results. In this way, *GPL* can achieve competitive test accuracy in CIFAR-10 sym 80% without co-training two mod-

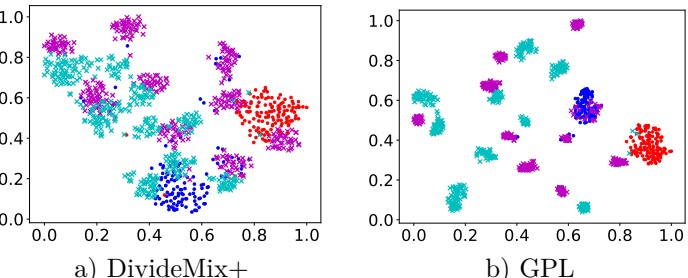

a) DivideMix+          b) GPL

Figure 3: Representations (t-SNE 2D embeddings) of two CIFAR-10 classes, 'cat' and 'truck', learned by DivideMix+ (left) and GPL (right), with 80% noise rate. Blue/red dots represent clean samples of cat/truck, while magenta and cyan crosses represent corrupted samples.

els. This finding is more prominent in CIFAR-100 asym 40%. **For (ii)**, we visualize feature representations of *GPL* and *DivideMix+* in 2-dimensional embeddings using t-SNE (der Maaten & Hinton, 2008). Figure 3 presents the normalised 2D embeddings of 500 randomly selected samples from each of two classes in CIFAR-10 sym 80%. We can see that both *GPL* and *DivideMix+* can accurately separate the clean instances of two classes (blue vs red). As for noisy instances (magenta and cyan), the representations learned by *GPL* are more fragmented and pushed away from clean clusters, showing that the SSL backbones of *GPL* is more effective in isolating noisy samples from clean ones. However, *DivideMix+* outperforms *GPL* in test accuracy because *DivideMix+* leverage two models for training and testing to improve accuracy.

### 6.4 Efficiency analysis

In Table 7, we study the efficiency of these two components by dividing all our instantiations into two parts and calculating their training time on CIFAR-10 sym 20% using an NVIDIA RTX 6000 GPU. We also break down the computation time per epoch (measured in seconds) for the SS process (Alg. 1, line 5) and SSL process (Alg. 1, line 6). **For the efficiency of SS**, *DivideMix+* is much faster than *DivideMix* because our per-minibatch SS strategy extracts and divides each minibatch of data faster. We can conclude that instantiations from our framework work more efficiently than simply combining SS and SSL components. *DivideMix-* is faster than *DivideMix*, which shows that *SPD* (used by *DivideMix-*) is faster than *GMM*. Moreover, *GPL* is much faster because only one network is trained. **For the efficiency of SSL**, *SPD-TE* is faster than *SPD-MM* and *SPD-PL* because the last two methods need to take an additional *MixUp* operation. The small time differences between these three instantiations indicate that various SSL backbones can be integrated into our framework efficiently.

Table 7: Comparison of training time on CIFAR-10.

| Method | GPL | DivideMix+ | DivideMix- | DivideMix | SPD-CE | SPD-TE | SPD-PL | SPD-MM |
|---|---|---|---|---|---|---|---|---|
| Total | 3.3 h | 8.2 h | 9.1 h | 9.4 h | 2.5 h | 3.0 h | 3.2 h | 3.6 h |
| SS/epoch | 9.2 s | 12.1 s | 15.5 s | 18.2 s | - | 37.2 s | 39.5 s | 44.1 s |

### 6.5 The Effectiveness and Efficiency of Two Networks

To study the effectiveness of training two networks on our instantiations, we compare six variants of our framework on CIFAR-10: DivideMix (1 network), DivideMix (2 networks), DivideMix+ (1 network), DivideMix+ (2 networks), GPL (1 network), and GPL (2 networks). "1/2 network/networks" means the instantiation without/with label co-refinement and co-guessing (Li et al., 2020b). The results of test accuracy and average training time are in Ta-

Table 8: Ablation Study of training with two networks on different instantiations on CIFAR-10.

| Model | Method | Sym-40% | Sym-80% | Asym-40% | Time |
|---|---|---|---|---|---|
| DivideMix | 1 network | 92.38 | 87.59 | 86.76 | 6.2 h |
| | 2 networks | **93.76** | **89.33** | **90.66** | 9.5 h |
| DivideMix+ | 1 network | 92.50 | 90.66 | 89.36 | 5.4 h |
| | 2 networks | **93.82** | **92.04** | **91.37** | 8.8 h |
| GPL | 1 network | 94.09 | 91.83 | 93.06 | 3.5 h |
| | 2 networks | **94.68** | **93.38** | **93.75** | 5.2 h |

ble 8. We observe that training with two networks provides a performance boost for the corresponding instantiations. The performance at the high noise rate is more significant than that of the medium noise rate. This is reasonable since the purpose of two diverged networks is to avoid error accumulation (Li et al., 2020b) during the training, a high noise rate will induce more errors for the single model.

## 7 Conclusion

This paper proposes a versatile framework called *SemiNLL* for NLL. This framework consists of two main parts: the mini-batch SS strategy and the SSL backbone. We conduct extensive experiments on benchmark-simulated and real-world datasets to demonstrate that *SemiNLL* can absorb a variety of SS strategies and SSL backbones, leveraging their power to achieve state-of-the-art performance in different noise scenarios. Moreover, we thoroughly analyze the effects of the two components in our framework. Recommendations on the choices of the components in our framework can be found in Appendix G. In future work, we hope to develop more advanced algorithms guided by this framework to tackle noisy labels.

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

**Appendix**

We provide a summary of contents below for easier navigation of the appendix of our paper titled "SemiNLL: A Framework of Noisy-Label Learning by Semi-Supervised Learning".

**Contents**

## A  Datasets

MNIST and FASHION-MNIST contain 60K training images and 10K test images of size $28 \times 28$. CIFAR-10 and CIFAR-100 contain 50K training images and 10K test images of size $32 \times 32$ with three channels. Clothing1M is a large-scale real-world dataset that consists of one million training images of size $224 \times 224$ from online shopping websites with labels annotated from surrounding texts. The estimated noise ratio is approximately 40% Xiao et al. (2015). Webvision Li et al. (2017) contains 2.4 million images crawled from the websites using the 1,000 concepts in ImageNet ILSVRC12Deng et al. (2009). Since the dataset is quite large, for quick experiments, we use the first 50 classes of the Google image subset. We test the trained models on the human-annotated WebVision validation set. The detailed characteristics of the datasets in the experiments are shown in Table 9.

Table 9: Summary of datasets used in the experiments.

|           | # of training | # of test | # of class | size |
|-----------|---------------|-----------|------------|------|
| *MNIST*     | 60,000        | 10,000    | 10         | $1 \times 28 \times 28$ |
| *F-MNIST*   | 60,000        | 10,000    | 10         | $1 \times 28 \times 28$ |
| *CIFAR-10*  | 50,000        | 10,000    | 10         | $3 \times 32 \times 32$ |
| *CIFAR-100* | 50,000        | 10,000    | 100        | $3 \times 32 \times 32$ |
| *Clothing1M* | 1,000,000    | 10,000    | 14         | $3 \times 224 \times 224$ |

## B  Network Structure

For MNIST, we use a simple 2-layer MLP following *Jocor* Wei et al. (2020). For FASHION-MNIST, we use a ResNet-18 He et al. (2016) following *GCE* Zhang & Sabuncu (2018). For CIFAR-10 and CIFAR-100, we use the "13-CNN" architecture Tarvainen & Valpola (2017), which is shown in Table 10. For Clothing1M, we use a ResNet-50 He et al. (2016) following *DivideMix* Li et al. (2020b). For mini WebVision, we use the inception-resnetv2 Szegedy et al. (2017).

Table 10: The "13-CNN" network architecture used in CIFAR-10 and CIFAR-100.

| Layer | Hyperparameters |
|-------|-----------------|
| Input | $32 \times 32$ RGB image |
| Convolutional | 128 filters, $3 \times 3$, *same* padding |
| Convolutional | 128 filters, $3 \times 3$, *same* padding |
| Convolutional | 128 filters, $3 \times 3$, *same* padding |
| Pooling | Maxpool $2 \times 2$ |
| Convolutional | 256 filters, $3 \times 3$, *same* padding |
| Convolutional | 256 filters, $3 \times 3$, *same* padding |
| Convolutional | 256 filters, $3 \times 3$, *same* padding |
| Pooling | Maxpool $2 \times 2$ |
| Convolutional | 512 filters, $3 \times 3$, *valid* padding |
| Convolutional | 256 filters, $1 \times 1$, *same* padding |
| Convolutional | 128 filters, $1 \times 1$, *same* padding |
| Pooling | Average pool ($6 \times 6 \rightarrow 1 \times 1$ pixels) |
| Softmax | Fully connected $128 \rightarrow 10$ (100) |

## C  Construction of Noisy Labels

The datasets used in our experiments are FASHION-MNIST, CIFAR-10, and CIFAR-100. To add noisy labels, we corrupt these datasets by two widely-used types of noisy labels: symmetric noise and asymmetric

noise (Li et al., 2020b; Zhou et al., 2021). Following these works, we manually corrupt the dataset by the label transition matrix $Q$, where $Q_{ij} = \mathrm{p}(\tilde{y} = j | y = i)$ given that noisy $\tilde{y}$ is flipped from clean $y$. For symmetric noise, we inject the symmetric label noise as follows:

$$
Q = \begin{bmatrix}
1 - \epsilon & \frac{\epsilon}{C-1} & \cdots & \frac{\epsilon}{C-1} & \frac{\epsilon}{C-1} \\
\frac{\epsilon}{C-1} & 1 - \epsilon & & & \frac{\epsilon}{C-1} \\
\vdots & & \ddots & & \vdots \\
\frac{\epsilon}{C-1} & & & 1 - \epsilon & \frac{\epsilon}{C-1} \\
\frac{\epsilon}{C-1} & \frac{\epsilon}{C-1} & \cdots & \frac{\epsilon}{C-1} & 1 - \epsilon
\end{bmatrix},
\tag{5}
$$

where $C$ is the number of classes, and $\epsilon$ is the noise ratio. Asymmetric noise is a well-known simulation of fine-grained classification with noisy labels, where annotators may make mistakes only within very similar classes. Its noise transition matrix $Q$ (taking 6 classes as an example) is obtained as follows:

$$
Q = \begin{bmatrix}
1 & 0 & 0 & 0 & 0 & 0 \\
0 & 1 - \epsilon & 0 & 0 & \epsilon & 0 \\
\epsilon & 0 & 1 - \epsilon & 0 & 0 & 0 \\
0 & 0 & 0 & 1 & 0 & 0 \\
0 & 0 & 0 & 0 & 1 & 0 \\
0 & 0 & \epsilon & 0 & 0 & 1 - \epsilon
\end{bmatrix}.
\tag{6}
$$

## D  Additional Experiment Results

### D.1  Results on More Real-world Datasets

We test two more real-world datasets, Food-101N (Lee et al., 2018) and ANIMAL-10N (Song et al., 2019). Food-101N is a dataset for food classification. It consists of 310,009 training images and 25,000 testing images in 101 classes collected from the web. The estimated label purity is 80%. We use ResNet-50 pre-trained on ImageNet and there are 30 epochs in total. ANIMAL-10N contains 10 animals with confusing appearances downloaded online. There are 50,000 training and 5,000 testing images. The estimated label noise rate is 8%. We use VGG-19 with batch normalization and there are 100 epochs in total. As shown in Table 11 and Table 12, we can conclude that our instantiation, DivideMix+, consistently outperforms all base methods, including its epoch-wise variant, DivideMix. GPL achieves competitive results using a single model.

Table 11: Test accuracy (%) on Food-101N.

| Method | Accuracy |
|---|---|
| Cross-Entropy | 81.53 |
| CleanNet (Lee et al., 2018) | 83.95 |
| DeepSelf (Han et al., 2019) | 85.10 |
| DivideMix (Li et al., 2020b) | 85.64 |
| **DivideMix+ (ours)** | **86.93** |
| GPL (ours) | 86.59 |

Table 12: Test accuracy (%) on ANIMAL-10N.

| Method | Accuracy |
|---|---|
| Cross-Entropy | 79.4 |
| Dropout (Srivastava et al., 2014) | 81.3 |
| SELFIE (Song et al., 2019) | 81.8 |
| DivideMix (Li et al., 2020b) | 82.4 |
| **DivideMix+ (ours)** | **83.6** |
| GPL (ours) | 83.2 |

### D.2  Sensitivity to the Batch Size of SS

We conduct a sensitivity analysis on the batch size of our two instantiations, DivideMix+ and GPL, on CIFAR-10 Sym 80%. In Table 13, we observe that larger the batch size achieves better test accuracy since extremely high noise mini-batch can be avoided. Too large batch size might result in performance degradation because the model might be trapped in local optimum (Loshchilov & Hutter, 2016).

Table 13: The test accuracy (%) of DivideMix+ and GPL with different batch sizes on CIFAR-10 Sym 80%.

| Batch size | 32 | 64 | 128 | 256 | 512 |
|---|---|---|---|---|---|
| DivideMix+ | 90.36 | 91.41 | 91.97 | **92.04** | 91.63 |
| GPL | 88.46 | 90.16 | 91.81 | **91.83** | 91.03 |

### D.3 Results of Different Model Architectures

We compared DivideMix, DivideMix+, and GPL on CIFAR-10 Sym-40%, Sym-80%, and Asym-40% noise with three different model architectures. We observe that DivideMix+ and GPL outperform DivideMix regardless of using deeper/wider networks. DivideMix+ outperforms GPL under high symmetric noise, while GPL achieves better results on low noise ratio and asymmetric noise.

Table 14: Comparison between DivideMix, DivideMix+, and GPL using different model architectures in test accuracy (%) on CIFAR-10. Key: WRN (Wide ResNet), PRN (PreActivation ResNet).

| Model | Method | Sym-40% | Sym-80% | Asym-40% |
|---|---|---|---|---|
| 13-CNN | DivideMix | 93.76 | 89.33 | 90.66 |
| | DivideMix+ | 93.82 | **92.04** | 91.37 |
| | GPL | **94.09** | 91.83 | **93.06** |
| PRN-18 | DivideMix | 93.02 | 90.94 | 91.60 |
| | DivideMix+ | 93.18 | **92.92** | 93.12 |
| | GPL | **94.64** | 90.92 | **93.44** |
| WRN-28 | DivideMix | 92.72 | 90.11 | 90.79 |
| | DivideMix+ | 93.26 | **92.57** | 92.06 |
| | GPL | **93.35** | 90.52 | **92.28** |

### D.4 Comparison between GPL (mini-batch) and GPL (epoch)

To demonstrate the effectiveness of our mini-batch SS, we also compare GPL (mini-batch) with GPL (epoch) with a batch size of 256 on CIFAR-10 in Table 15. We observe that the test accuracy of GPL (mini-batch) constantly outperforms GPL (epoch).

Table 15: The test accuracy (%) of GPL (epoch) and GPL (mini-batch) CIFAR-10.

| Method | Sym-40% | Sym-80% | Asym-40% |
|---|---|---|---|
| GPL (epoch) | 93.13 | 88.96 | 91.55 |
| GPL (mini-batch) | **94.09** | **91.83** | **93.06** |

## E Limitations

**Mini-batch SS.** Noise ratios in different mini-batches inevitably fluctuate since each mini-batch of data is randomly sampled from the whole dataset. When the overall noise level is high and the batch size is too small, the SS strategy might have trouble distinguishing clean and noisy samples in some severely corrupted mini-batches, which may deteriorate the overall performance. For example, in DivideMix+, we apply GMM to each mini-batch of data. The fluctuation of the noisy ratio in each mini-batch might result in inconsistent data selection criteria (clean threshold). Moreover, we conduct a sensitivity analysis on the batch size of our two instantiations, DivideMix+ and GPL, on CIFAR-10 Sym 80% in Appendix D.2.

**SSL component.** Under the symmetric noise, the labels are uniformly corrupted. If the SS strategy can distinguish clean/noisy samples well enough, the class distribution in the labeled/unlabeled sets is uniform. While under the asymmetric noise, the labels are flipped to similar classes, resulting in the class-imbalanced labeled/unlabeled data. However, traditional SSL methods (Berthelot et al., 2019; Arazo et al., 2020; Sohn et al., 2020) (including the SSL backbones of our instantiations) assume that the class distribution of labeled and/or unlabeled data is balanced. In Table 1 (symmetric noise), methods using SSL (DivideMix, DivideMix+, and GPL) outperform others in most cases. In Table 2(asymmetric noise), methods using SSL perform slightly worse than other SOTA methods in MNIST and FASHION MNIST. Please be noted that our framework can absorb the advantages of various SS and SSL methods. To deal with the class-imbalanced issue in asymmetric noise, we can develop new instantiations using SSL methods designed to tackle class-imbalance data (Kim et al., 2020; Wei et al., 2021a).

## F   Broader Impact

If the clean examples are incorrectly identified, the network could inadvertently ignore meaningful labels, and vice versa, i.e., the network could learn from meaningless noise rather than clean labels. This motivates a broad discussion about potentially catastrophic effects. We formalize a versatile framework that leverages SS to select clean samples and SSL to fully use the noisy samples by removing their labels. And we believe it will substantially impact labor-intensive jobs of checking data label quality, such as training models from the web-crawled images (Schroff et al., 2010) and medical data analysis (Miotto et al., 2018). One potential risk caused by the wrong sample selection process is the increased chances of the model over-fitting potential outliers in the data that may lead to erroneous or misleading results, i.e., the misdiagnosing of patients. In the future, we will develop more precious and safer SS strategies to alleviate this negative impact.

## G   Choices of the Components in Our Framework

The components considered in our framework are: SS $\in$ {GMM, SPD}, SSL $\in$ {Temporal Ensembling, MixMatch, Pseudo-Labeling}, SS scope $\in$ {Mini-batch, Epoch}, number of networks $\in$ {1, 2}. We conduct a more systematic comparison by considering all the components mentioned above. We also make some recommendations based on the comparison.

### G.1   Choices of SS Scope

**Analysis.** The advantages and limitations of mini-batch SS compared to epoch-wise SS are thoroughly discussed in Section 3.1 and Appendix E, respectively.

**Empirical evidence.** In Section 6.2, we demonstrated the benefit of mini-batch-wise SS over epoch-wise SS by comparing DivideMix and DivideMix+. In Appendix D.4, we also compare GPL (mini-batch) with GPL (epoch) on CIFAR-10. A sensitivity analysis of the batch size of mini-batch SS is conducted in Appendix D.2.

**Our recommendation.** If we choose a bigger batch size, our mini-batch-wise SS can be effective for both symmetric and asymmetric noise. From the sensitivity analysis on the batch size, the batch size needs to be bigger than 64 to avoid the fluctuation of the noisy ratio in each mini-batch. Thus, in the following comparisons, we only use mini-batch-wise SS with a batch size of 256.

### G.2   Choices of Number of Networks

**Analysis.** The purpose of two diverged networks is to avoid error accumulation (Li et al., 2020b) during the training. Under high noise rates, more errors will be induced during the training of a single model. Thus, training with two diverged networks is more effective in severely corrupted datasets. The drawback is the increase in training time and computational resources.

**Empirical evidence.** In Section 6.5, we demonstrated the effectiveness and efficiency of training with two networks.

**Our recommendation.** Using two networks can boost performance for the corresponding instantiations, especially at high noise rates. The shortcoming of using two networks is the increase in training time. So it might not be necessary at low noise rates or on simple datasets (MNIST). In the following comparisons on CIFAR-10, we will use two networks for all instantiations for a fair comparison.

### G.3 Choices of Different SS and SSL Combinations

The methods considered in this work are SS ∈ {GMM, SPD}, SSL ∈ {Temporal Ensembling, MixMatch, Pseudo-Labeling}. We combine each SS and each SSL method to conduct a systematic comparison in Table 16. We denote these instantiations as "SS" + "SSL". Please be noted that mini-batch SS and two networks are used in all the instantiations according to previous recommendations.

Table 16: The test accuracy (%) of instantiations of different SS and SSL methods under high/low symmetric noise in CIFAR-10.

| Noise type | SS\SSL | Temporal Ensembling | MixMatch | Pseudo-Labeling |
|---|---|---|---|---|
| Sym-40% | GMM | 85.79 | 93.82 | **94.68** |
| | SPD | 84.60 | 92.68 | 94.33 |
| Sym-80% | GMM | 54.37 | 92.04 | **93.38** |
| | SPD | 51.56 | 90.98 | 92.20 |
| Asym-40% | GMM | 64.53 | 91.37 | **93.75** |
| | SPD | 68.74 | 92.11 | 92.14 |

We discovered some general rules and synergistic pairs when combining different SS and SSL methods using our framework:

(i) General rules regarding SS: Under symmetric noise, instantiations with GMM achieve higher test accuracy than instantiations with SPD, while the results are reversed under asymmetric noise. This is because GMM distinguishes clean and noisy samples based on their loss distribution. For asymmetric noise, most samples have near-zero normalized loss due to the low entropy predictions of the network (Li et al., 2020b), causing performance deterioration for GMM. Since SPD leverages the prediction of its network to choose clean samples, its performance is more sensitive to noisy rates rather than noise type.

**Our recommendation.** For symmetric noise and high noise ratios, GMM should be used as the SS strategy. For asymmetric noise, SPD should be used as the SS strategy. For low noise ratios, SPD might be more computationally efficient (Table 7) in achieving competitive test accuracy.

(ii) General rules regarding SSL: Instantiations with Temporal Ensembling are much worse than those with MixMatch and Pseudo-Labeling because Temporal Ensembling only uses consistency regularization for unsupervised loss. In general, instantiations with Pseudo-Labeling achieve higher accuracy than the other two because the regularization term used in Pseudo-Labeling prevents the model from assigning all labels to a single class at the early training stage. From the results of Asym 40%, we observe performance degrade of all SSL methods, especially Temporal Ensembling, combined with the same SS method (even though SPD can alleviate the harm of asymmetric noise). This is because the class distribution in the labeled/unlabeled sets after the SS process is class-imbalanced under the asymmetric noise. However, all the methods considered in this paper assume that the class distribution of labeled and/or unlabeled data is balanced.

**Our recommendation.** Our guidance is to select a suitable SSL method for the corresponding labeled/unlabeled sets after the SS process instead of "using the strongest SSL algorithm" to achieve the best test accuracy. For example, if the class distribution of labeled/unlabeled sets is (close to) balanced, we can choose more advanced mainstream SSL methods (Sohn et al., 2020; Zhang et al., 2021) as long as we take into account computational resources and efficiency. If labeled/unlabeled sets are class-imbalanced, we can choose SSL methods specifically designed to tackle class-imbalance data (Kim et al., 2020; Wei et al., 2021a).

(iii) Synergistic combination: We surprisingly find that GMM + Pseudo-Labeling, the two-network version of the original GPL, achieves the best performance in all settings, including the asymmetric noise where SPD is supposed to be stronger.

**Our recommendation.** GMM + Pseudo-Labeling achieves the best performance for both symmetric and asymmetric noise. We believe its superior performance benefits from the conceptually simple idea of Pseudo-Labeling to reduce confirmation bias generated from the SS process. Please be noted that Pseudo-Labeling was not published at top conferences. That being said, we will keep exploring the possibility of more combinations of SS and SSL based on our framework to figure out the chemistry in between, instead of just simply piling up SOTA SS and SSL methods published in top conferences.

## H    Illustrations

To better understand the differences between *DivideMix-*, *DivideMix*, and *DivideMix+*, their illustrations are shown in Figure 4 (a), (b), and (c), respectively.

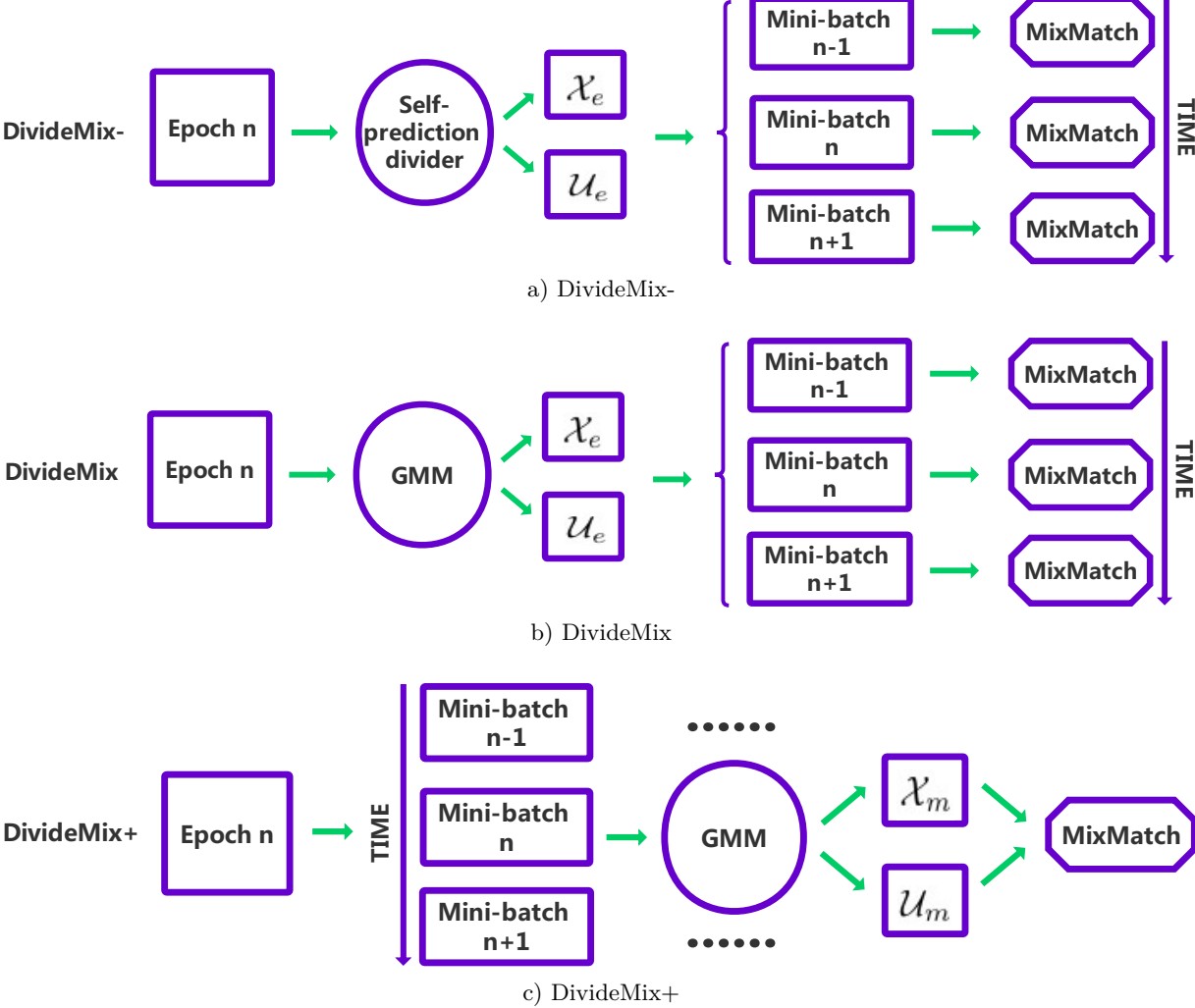

Figure 4: Comparisons between: (a) *DivideMix-*, (b) *DivideMix*, and (c) *DivideMix+*. Squares represent data. Circles represent SS strategy. Octagons represent SSL backbone.

