# OpenReview forum: "SemiNLL: A Framework of Noisy-Label Learning by Semi-Supervised Learning"
_TMLR — Accepted by TMLR_

### Review · Reviewer_TbTQ · 2022-06-10

**Summary Of Contributions:**

This paper proposes SemiNLL, a versatile framework for noisy label learning. It combines the advantages of sample selection strategy and semi-supervised learning in a better manner. The results show promising performance.

**Broader Impact Concerns:**

No Broader Impact concerns.

**Requested Changes:**

Please refer to the weakness part.

The authors may analyze the framework in more detail and show the mini-batch-wise usage of SS and SSL is more important than the epoch-wise usage of SS and SSL.

More real-world datasets should be compared.

The proposed method may depend on the assumptions of SSL. On which types of noisy learning datasets the method works well should be discussed.

**Strengths And Weaknesses:**

The authors revisit noisy label learning based on their usage of the Sample Selection (SS) strategy and Semi-Supervised Learning (SSL). A new framework to combine these two important components is proposed. The main idea of the framework is simple, and the authors claim the importance of the mini-batch sample selection as well as the SSL backbones. Various combinations of SS and SSL are investigated in the paper. The authors also compare the proposed method with various methods on synthetic and real datasets.

Weakness:
1. The main idea of the paper is not clear after reading the introduction.
2. The proposed SemiSLL alternatively applies SS and SSL to update the model, whose effectiveness is only supported by some empirical results. The importance of the change should be discussed in more detail.
3. The authors compare SemiSLL with previous methods in Section 3, but the possible drawbacks of the method are not fully discussed. For example, the mini-batch sample selection can mitigate the confirmation bias, but will make the SS more difficult with limited batch-size, since some methods need a holistic evaluation of data to determine the clean and noisy data.

---

> ### Author Response · Authors · 2022-06-26
> **Response to Reviewer TbTQ (1/2)**
>
> Thank you for your constructive comments. We address all of your concerns point by point as below:
>
> ### **C1** *The main idea of the paper is not clear after reading the introduction.*
>
> We will briefly describe our motivation, method, and contributions stated in the introduction.
>
> - **Motivation**: A typical strategy in noisy label learning called **sample selection (SS)** selects only clean samples for training. However, those unselected samples are discarded without further exploitation. Recent methods leverage **semi-supervised learning (SSL)** to use all given samples fully.
> - **Method**:  We propose **a versatile framework** that investigates how to naturally combine various SS and SSL components based on their effects and efficiencies.
> - **Contributions**: Our framework is valuable because it can not only provide an **important prototype** in the NLL community for further exploration into SS and SSL but can also act as a **conclusive work** to prevent future researchers from simply reinventing the wheel.
>
> We have revised the introduction accordingly.
>
> ### **C2** *The authors may analyze the framework in more detail and show the mini-batch-wise usage of SS and SSL is more important than the epoch-wise usage of SS and SSL*.
>
> Please be noted that the **difference between our mini-batch-wise framework and previous epoch-wise methods** lies in the process where clean/noisy samples are divided. This results in the alternate process of SS and SSL between consecutive mini-batches. We give a detailed analysis of the advantages of our mini-batch-wise SS and SSL over epoch-wise one:
>
> - **Avoid confirmation bias**. In the case of epoch-wise SS, the divided clean/noisy sets are incorporated into the SSL phase and will not be updated till the next epoch. Thus, the confirmation bias induced from those wrongly divided samples will accumulate within the whole epoch. Our mini-batch SS strategy divides each mini-batch of samples into clean/noisy batches right before updating the network using SSL backbones. In the next mini-batch, the updated network can know better to distinguish clean and noisy samples, alleviating the confirmation bias mini-batch by mini-batch.
>
> - **Improve computational efficiency**. Since the time complexity of most SS methods (including GMM and self-prediction divider) is not linear, the number of operations increases dramatically as the input size increases. Table 7 compares the training time of DividMix+ (mini-batch-wise) and DivideMix (epoch-wise) on CIFAR-10, showing DivideMix+ is more computationally efficient than DivideMix in both the SS process and the whole training process.
>
> - **Inject stochasticity**. During the epoch-wise SS process, the model tends to select the confident samples that have been selected in previous epochs due to the model overfitting their labels. In this way, some confident but noisy samples will keep being selected by the model, resulting in performance degradation. The mini-batch-wise SS can inject stochasticity in training since each mini-batch of data is randomly sampled from the whole dataset, avoiding the model constantly selecting the same confident samples.
>
> - **Up-to-date model for SS**. The mini-batch-wise usage of SS and SSL makes the data selection up-to-date. The model used to select the clean samples is updated using the SSL method at each minibatch. In comparison, the epoch-wise usage of the SS process selects the clean samples based on the model trained by SSL from the last epoch.
>
> We have uploaded a new version of the paper that contains the advantages of mini-batch-wise SS in Section 3.1.
>
> ### **C3** *More real-world datasets should be compared*.
> Please refer to our reply to Q2 in the general response.

---

> > ### Author Response · Authors · 2022-06-26
> > **Response to Reviewer TbTQ (2/2)**
> >
> > ### **C4** *The possible drawbacks of the method are not fully discussed.....Mini-batch operation will make the SS more difficult with limited batch size, since some methods need a holistic evaluation of data to determine the clean and noisy data*.
> > Thank you for pointing this drawback out. Indeed, noise ratios in different mini-batches inevitably fluctuate since each mini-batch of data is randomly sampled from the whole dataset. When the overall noise level is high and the batch size is too small, the SS strategy might have trouble distinguishing clean and noisy samples in some **severely corrupted mini-batches**, which may deteriorate the overall performance. For example, in DivideMix+, we apply GMM to each mini-batch of data. The fluctuation of the noisy ratio in each mini-batch might result in inconsistent data selection criteria (clean threshold). Moreover, we conduct a **sensitivity analysis on the batch size of mini-batch SS** of our two instantiations, DivideMix+ and GPL, on CIFAR-10 Sym 80%.
> >
> > | Method\Batch size | 32 | 64 | 128 | 256 | 512 |
> > | :----| :----: | :----: | :----: | :----: | :----: |
> > | DivideMix+| 90.36 | 91.41 | 91.97 | **92.04** | 91.63 |
> > | GPL | 88.46 | 90.16 | 91.81 | **91.83** | 91.03 |
> >
> > We observe that larger the batch size achieves better test accuracy since extremely high noise mini-batch can be avoided. If we choose **a bigger batch size**, our **mini-batch-wise SS** can be more effective than epoch-wise SS. From the sensitivity analysis on the batch size, the **batch size needs to be bigger than 64** to avoid the fluctuation of the noisy ratio in each mini-batch. Too large batch size might result in performance degradation because the model might be trapped in local optimum [1].
> >
> > We have uploaded a new version of the paper that contains the limitations of the mini-batch SS and the sensitivity analysis in Appendix F and E.2, respectively.
> >
> >
> > ### **C5** *The proposed method may depend on the assumptions of SSL. On which types of noisy learning datasets the method works well should be discussed*.
> > Thank you for this suggestion! Under the symmetric noise, the labels are uniformly corrupted. If the SS strategy can distinguish clean/noisy samples well enough, the class distribution in the labeled/unlabeled sets is uniform. While under the asymmetric noise, the labels are flipped to similar classes, resulting in the class-imbalanced labeled/unlabeled data. However, **traditional SSL methods** [2, 3, 4] **(including the SSL backbones of our instantiations) assume that the class distribution of labeled and/or unlabeled data is balanced**. In Table 1 (symmetric noise), methods using SSL (DivideMix, DivideMix+, and GPL) outperform others in most cases. In Table 2(asymmetric noise), methods using SSL perform slightly worse than other SOTA methods in MNIST and FASHION MNIST. Please be noted that our framework can absorb the advantages of various SS and SSL methods. **To deal with the class-imbalanced issue in asymmetric noise, we can develop new instantiations using SSL methods designed to tackle class-imbalance data [5, 6]**.
> >
> > We have uploaded a new version of the paper that contains the limitations of the SSL component in Appendix F.
> >
> > [1] SGDR: Stochastic Gradient Descent with Warm Restarts, ICLR 2017
> >
> > [2] Pseudo-labeling and confirmation bias in deep semi-supervised learning, IJCNN 2020
> >
> > [3] Mix-match: A holistic approach to semi-supervised learning, NeurIPS 2019
> >
> > [4] Fixmatch: Simplifying semi-supervised learning with consistency and confidence, NeurIPS 2020
> >
> > [5] CReST: A class-rebalancing self-training framework for imbalanced semi-supervised learning, CVPR 2021
> >
> > [6] Distribution aligning refinery of pseudo-label for imbalanced semi-supervised learning, NeurIPS 2020

---

### Review · Reviewer_8Gpa · 2022-06-16

**Summary Of Contributions:**

This paper proposes a framework called SemiNLL  that integrates SS and SSL to improve accuracy of deep learning models. The SemiNLL was examined on Mnist, Fashion Mnist, Cifar-10 and 100 datasets for image classification tasks. In the analyses, the proposed SemiNLL performs on par with, or slightly worse/better compared to the state-of-the-art.


**Broader Impact Concerns:**

N/A.

**Requested Changes:**

Some minor issues:

The abstract should be clarified. There are two major parts to be improved. First, describe the main novelty; that is, in “naturally combine different SS and SSL components”, please briefly describe “natural combination” and components. Second, please provide main results, i.e. describe what “the new state-of-the-art” is.

Some of the requested changes were provided in the previous section.

Briefly:

(1) Several parts and statements of the paper should be clarified and more precise definitions should be provided.

(2) The claims should be revised and updated.

(3) Additional theoretical and experimental results should be given to support the claims on the novelty, if the paper is considered as an analysis paper. If authors consider that the main contribution of the paper is algorithmic, then novelty of the proposed framework should be improved by a novel method integrating SS and SSL. For the moment, it is not clear how "natural" their combination is.

(4) The superiority of the proposed method over state-of-the-art is not clear. Additional analyses with different larger scale datasets and models should be provided for justification.



**Strengths And Weaknesses:**

Strengths:

The proposed pipeline is straightforward.

Weaknesses:

Major problems with the paper are over-claims, unclear definitions and methods, and limited novelty:

(1) Definition of noisy labels: In the paper, noisy labels are defined as unlabelled sets. For instance, it is stated that “Another recent method, DivideMix (Li et al., 2020), leverages Gaussian Mixture Model (GMM) (Permuter et al., 2006) to distinguish clean (labeled) and noisy (unlabeled) data, and then uses a strong SSL backbone called MixMatch (Berthelot et al., 2019).”

However, noisy labels can be also defined by inducing noise to the labels, while the paper simply considers unlabelled data (according to the statements). Please define “a noisy label” more precisely, and update the related parts accordingly.

(1.a) Following the above issue, it is not clear how “noisy samples” are selected or constructed in the experimental analyses.

(1.b) The paper proposes 2 methods for SS.

First, GMM based methods.

Second: “The self-prediction divider which determines potentially clean samples in a mini-batch if the samples’ maximal likelihood predictions of the network match their annotated labels. Specifically, the samples are divided into the labeled set only if the model predicts the annotated label to be the correct class with the highest likelihood.”

However, how these GMM and max-likelihood estimation models are trained and employed are not explained. They are just defined as black-boxes which split data into clean labelled /noisy (unlabelled?) sets.

(1.c) Novelty of SS: The main difference between the proposed SS and related work is mini-batch sample selection. It is not clear why this method should work compared to the other baseline. Indeed, in the analyses, this method performs on par in various setups.

(2) SSL: The paper briefly ensembles different SSLs with different SSs, and experimentally analyze their accuracy. However, stronger experimental and theoretical analyses are required to justify the results. In addition, it is claimed that they are “naturally combined”, however, this claim is vague and should be precisely supported.

(3) The paper claims that the proposed method "outperform other state-of-the-art noisy-label learning methods." However, in the analyses, various different combinations of cascaded integration of SSs and SSLs are given, and the proposed methods perform on par with, or slightly worse/better compared to the state-of-the-art. To resolve this, first experimental results should be described more precisely revising the claim. Second, additional analyses should be performed.

---

> ### Author Response · Authors · 2022-06-26
> **Response to Reviewer 8Gpa (1/3)**
>
> Thank you for your constructive comments. We address your concerns point by point.
>
> ### **C1** *Definition of noisy labels: In the paper, noisy labels are defined as unlabelled sets*.
> With all due respect, **the noisy labels in this paper are defined as the noise in training set labels instead of the unlabelled sets**. In DivideMix, GMM is used to divide the original dataset into the clean set and the noisy set. The labels of the clean sets will be kept, while the labels of the noisy set are discarded, turning into an unlabeled set. Thus, the problem setup becomes an SSL task. We revise the statement of DivideMix accordingly to avoid confusion. And we give a clear definition of "noisy labels" in the introduction.
>
> ### **C2** *It is not clear how “noisy samples” are selected or constructed in the experimental analyses*.
> In Section 5.1, we described two types of noisy labels, **symmetric noise and asymmetric noise, commonly used in the noisy label learning community** [1,2,3]. We have added the corresponding label transition matrix Q in Appendix C.
>
> ### **C3** *How these GMM and max-likelihood estimation models are trained and employed are not explained*.
> Thanks for raising this concern! We will explain how GMM and max-likelihood estimation models (SPD) are trained and employed, respectively.
>
> ```
> GMM
> ```
> Since DNNs tend to learn simple patterns first before fitting noisy samples [11], clean/noisy samples have lower/higher loss during the early epochs of training. Therefore, we can determine whether a sample is more likely to be clean or noisy based on its loss value. Given a classification loss $\ell_{i}$ (e.g., cross entropy) of sample $(x_{i}, y_{i})$ in the dataset, we calculate the probability density function (pdf) of a mixture model of k (in our case k = 2) components on the loss $\ell_{i}$ using Expectation-Maximization technique:
>
> $p\!\left(k\mid\ell_{i}\right)=\frac{p\!\left(k\right)p\!\left(\ell_{i}\mid k\right)}{p\!\left(\ell_{i}\right)}$
>
> where $k=0\left(1\right)$ denotes the clean (noisy) set, and $p\!\left(\text{clean}\mid\ell_{i}\right)$ is the posterior of the smaller-mean (clean) component of the GMM. The clean set $\mathcal{X}$ and $\mathcal{Y}$ the noisy set can be defined in Eq. (4) in Appendix D.
> ```
> SPD
> ```
> Based on the phenomenon that DNN’s predictions tend to be consistent on clean samples and inconsistent on noisy samples in different training iterations [10], we select the correctly annotated samples via the consistency between the original label set and the model’s own predictions. If the model predicts the class with the highest likelihood to be the annotated label, the sample $(x_{i}, y_{i})$ is divided into the clean set. Otherwise, the sample is divided into the noisy set. Please kindly refer to Eq. (5) and Eq. (6) in Appendix D for a better view of the formulas.
>
> Due to the organization of Section 4 and limited space, we have added the explanations of GMM and max-likelihood estimation models (SPD) in Appendix D.
>
> ### **C4** *The main difference between the proposed SS and related work is mini-batch sample selection. It is not clear why this method should work compared to the other baseline.*.
> Thanks for your constructive comments! Please refer to our reply to Q1 in the general response.

---

> > ### Author Response · Authors · 2022-06-26
> > **Response to Reviewer 8Gpa (2/3)**
> >
> > ### **C5** *Additional theoretical and experimental results should be given to support the claims on the novelty, if the paper is considered as an analysis paper. If authors consider that the main contribution of the paper is algorithmic, then novelty of the proposed framework should be improved by a novel method integrating SS and SSL. For the moment, it is not clear how "natural" their combination is*.
> >
> > Thank you for your constructive comments! We will address your concerns point by point.
> >
> > - **Novelty of analysis paper**: We agree the novelty of qualified research should not be limited to algorithm design only. One should also consider the novelty of framework design and its contributions to the research community. For example, GraphSAGE [4] proposed a simple yet universal framework for inductive learning on graph data, on which hundreds of papers were built. Like GraphSAGE, this paper proposes **a versatile framework** to investigate how to combine different SS and SSL components, which has **never been studied explicitly** in the noisy label learning community. Based on our framework, one can not only understand different SS and SSL components better, but more synergistic combinations can also be created. This can be done according to the performance and efficiency of each component. Many recent preprints [5, 6] could be taken as special instantiations of our framework, which indicates that a **conclusive analysis work** like this paper is vitally necessary to prevent future researchers from reinventing the wheel.
> >
> > - **Meaning of "natural combination"**: First, we will explain it **from the perspective of our framework**. Sample Selection (SS) [3, 7, 8, 9] has been widely used in noisy label learning to select clean samples for training. However, those noisy samples are discarded without further exploiting the associated image contents. Thus, it is a **natural** operation to remove the labels of these noisy samples and leverage semi-supervised learning (SSL) to learn from them as the unlabeled samples. In this way, all clean/noisy samples will be fully used to improve the performance of the model. Moreover, **we have added a more systematic and detailed analysis of the combinations of the components based on our framework** in Appendix G.3. The components considered in this work are SS = {GMM, SPD} and SSL = {Temporal Ensembling, MixMatch, Pseudo-Labeling}. We combine each SS and each SSL method to conduct a systematic comparison. We denote these instantiations as "SS" + "SSL" and record their test accuracy on CIFAR-10 Sym 40%, Sym 80%, and Asym 40%. We give a brief conclusion here and leave a detailed analysis to the newly added Appendix G.3: (i) For symmetric noise and high noise ratios, **GMM** should be used as the SS strategy. For asymmetric and low noise ratios, **SPD** should be used. (ii) If the class distribution of labeled/unlabeled sets after the SS process is **balanced**, we can choose more advanced **mainstream SSL** methods. If labeled/unlabeled sets are **class-imbalanced**, we can choose SSL methods **specifically designed** to tackle class-imbalance data. (iii) **GMM + Pseudo-Labeling** achieves the best performance for both symmetric and asymmetric noise, indicating the combination of GMM and Pseudo-Labeling is **more synergistic**.
> >
> > - **Additional theoretical results**: Thank you for this comment. And we admit that our framework lacks adequate theoretical results. However, we would argue that **our framework is the first to explicitly study the combination of SS and SSL in the NLL community, and the revised version is well supported by empirical results and thorough analysis**. This research pattern could also be found in many methods in the NLL community (which also fits the scope of TMLR). For instance, SELF [10] proposed a noise filtering mechanism based on the Mean-Teacher model and its self-ensemble prediction without any theoretical support. DivideMix [2] leverages an ensemble of two networks to divide and relabel clean/noisy samples purely based on empirical results. Both of them could be taken as special instantiations of our framework. From the experimental results and corresponding analysis, our framework can absorb various SS strategies and SSL backbones, utilizing their traits to tackle different noisy label settings. Therefore, we believe our framework is **valuable** to the NLL community despite lacking adequate theoretical results. **We also humbly agree that theoretical results are important**, and we will provide theoretical guarantees in future works.
> >
> > - **Additional experiemntal results**: The components considered in our framework are: SS = {GMM, SPD}, SSL = {Temporal Ensembling, MixMatch, Pseudo-Labeling}, SS scope = {Mini-batch, Epoch}, number of networks = {1, 2}. We conduct a more systematic comparison by considering all the components mentioned above. We also make some recommendations based on the comparison. Please refer to Appendix G for more details.

---

> > > ### Author Response · Authors · 2022-06-26
> > > **Response to Reviewer 8Gpa (3/3)**
> > >
> > > ### **C6** *The superiority of the proposed method over state-of-the-art is not clear. Additional analyses with different larger-scale datasets and models should be provided for justification*.
> > > Thank you for this valuable suggestion!
> > >
> > > - **Analysis of experimental results**: We have revised Section 5.2 accordingly by more precisely describing the performance of DivideMix+ and GPL compared to other methods.
> > >
> > > - **Different larger-scale datasets**: Please refer to our reply to Q2 in the general response.
> > >
> > > - **Different models**: We compared DivideMix, DivideMix+, and GPL on CIFAR-10 Sym-40%, Sym-80%, and Asym-40% noise with three different model architectures: 13-CNN, WRN (Wide ResNet), PRN (PreActivation ResNet). We observe that DivideMix+ and GPL outperform DivideMix **regardless of using deeper/wider networks**. DivideMix+ outperforms GPL under high symmetric noise, while GPL achieves better results on low noise ratio and asymmetric noise. We have added the corresponding results in Appendix E.3.
> > >
> > > 	| Method | Method | 32 | 64 | 128 |
> > > 	| :----| :----: | :----: | :----: | :----: |
> > > 	| 13-CNN | DivideMix | 93.76 | 89.33 | 90.66 |
> > > 	| 13-CNN | DivideMix+ | 93.82 | **92.04** | 91.37 |
> > > 	| 13-CNN | GPL | **94.09** | 91.83 | **93.06** |
> > > 	| PRN-18 | DivideMix | 93.02 | 90.94 | 91.60 |
> > > 	| PRN-18 | DivideMix+ | 93.18 | **92.92** | 93.12 |
> > > 	| PRN-18 | GPL | **94.64** | 90.92 | **93.44** |
> > > 	| WRN-28 | DivideMix | 92.72 | 90.11 | 90.79 |
> > > 	| WRN-28 | DivideMix+ | 93.26 | **92.57** | 92.06 |
> > > 	| WRN-28 | GPL | **93.35** | 90.52 | **92.28** |
> > >
> > >
> > > ### **C7** *The abstract should be clarified*.
> > > Thank you for pointing this out. We have revised the abstract accordingly.
> > >
> > >
> > > [1] Generalized cross entropy loss for training deep neural networks with noisy labels, NeurIPS 2018
> > >
> > > [2] DivideMix: Learning with Noisy Labels as Semi-supervised Learning, ICLR 2020
> > >
> > > [3] Combating noisy labels by agreement: A joint training method with co-regularization, CVPR 2020
> > >
> > > [4] Inductive Representation Learning on Large Graphs, NeurIPS 2017
> > >
> > > [5] LongReMix: Robust Learning with High Confidence Samples in a Noisy Label Environment, https://arxiv.org/pdf/2103.04173.pdf
> > >
> > > [6] Dst: Data selection and joint training for learning with noisy labels, https://arxiv.org/pdf/2103.00813.pdf
> > >
> > > [7] Co-teaching: Robust training of deep neural networks with extremely noisy labels, NeurIPS 2018
> > >
> > > [8] How does disagreement help generalization against label corruption? ICML 2019
> > >
> > > [9] Sample selection with uncertainty of losses for learning with noisy labels, ICLR 2022
> > >
> > > [10] Self: Learning to filter noisy labels with self-ensembling, ICLR 2020
> > >
> > > [11] A closer look at memorization in deep networks, ICML 2017

---

### Review · Reviewer_e8Ug · 2022-06-17

**Summary Of Contributions:**

This paper proposes a framework called SemiNLL for performing deep learning in the presence of noisy labels. SemiNLL has two main components: (1) a sample selection (SS) algorithm which predicts whether examples are clean or noisy, and (2) a semi-supervised learning (SSL) algorithm that infers new labels for the noisy examples. The key benefit of SemiNLL is its modularity since a variety of SS or SSL algorithms can be used in the framework. Two main instantiations of SemiNLL are investigated: DivideMix+ (GMM + MixMatch with mini-batch sample selection) and GPL (GMM + PseudoLabel). Experiments are conducted on a variety of image datasets with both symmetric and asymmetric noise and the proposed instantiations were shown to have SOTA performance.

Contributions include:
* Formalization of a framework called SemiNLL for noisy label learning that combines SS and SSL.
* Both the DivideMix+ and GPL instantiations are shown to have very strong accuracy for both symmetric and asymmetric noise.
* Demonstration of the benefit of mini-batch SS vs. epoch-level SS (DivideMix+ vs. DivideMix).

**Broader Impact Concerns:**

Regarding the use of SS and SSL: if the clean examples are incorrectly identified then the network could inadvertently ignore labels that actually are meaningful, and vice versa, i.e. the network could learn from meaningless noise rather than clean labels. This could have catastrophic effects depending on the application. A discussion of these potential risks should be added.

**Requested Changes:**

Each of the following are important:
* Add a more systematic comparison of instantiations of SemiNLL. Of the components considered in this work, from what I gather, SS = {GMM, SPD}, SSL = {MixMatch, PseudoLabel, Temporal Ensembling}, SS scope = {Mini-batch, Epoch}, # of networks = {1, 2}. Only a subset of these possibilities are considered. The chosen instantiations, in particular DivideMix+ and GPL, do indeed achieve good performance but the work would have greater impact if a more systematic comparison can be made. Of these, comparing 1 vs. 2 networks seems to be the most important since as mentioned above the comparison with respect to GPL is currently an apples to oranges comparison.
* Add discussion that makes a recommendation, based on empirical evidence, for how future researchers should go about selecting SS and SSL algorithms within the SemiNLL framework.
* Add discussion of limitations of the proposed approach.

**Strengths And Weaknesses:**

Strengths
* The proposed SemiNLL framework is intuitive and easy to comprehend.
* A breadth of datasets, noise levels, and baselines were used in the experimental comparisons.
* Writing is good quality.

Weaknesses
* The experimental comparison among instantiations of SemiNLL is not comprehensive. Not all combinations of the components were tried, and so the takeaways of the experiments are somewhat unclear. The most clear is the benefit of mini-batch vs. epoch-level SS. Beyond this, the conclusions become more unclear. See e.g. Sec. 6.3 which makes the point that GPL is more effective in separating noisy samples, yet DivideMix+ has better performance due to it having two models. However, in Sec. 4.2 it was stated that training two networks is out of scope. Yet this choice leads prevents an apples to apples comparison from being made.
* The message about how to choose specific SS and SSL algorithms is unclear. Currently it seems that the guidance is to use the strongest SS and the strongest SSL algorithm. Are certain pairings of SS and SSL algorithms more synergistic than others, i.e. the sum is greater than the parts? Addressing questions like this are important to prevent future research from "reinventing the wheel", which was stated as a goal of the work.
* No discussion of limitations of the proposed method. One point in particular would be to discuss the effect of batch size on the mini-batch SS.

---

> ### Author Response · Authors · 2022-06-26
> **Response to Reviewer e8Ug (1/3)**
>
> Thanks for your valuable comments, and we are glad that you appreciate the contributions of our framework and instantiations. Below are the responses to your comments/questions.
>
> ### **C1** *The work would have greater impact if a more systematic comparison can be made. Add discussion that makes a recommendation, based on empirical evidence, for how future researchers should go about selecting SS and SSL algorithms within the SemiNLL framework*.
> Thank you for these valuable suggestions! We conduct **a more systematic comparison** that includes SS, SSL, SS scope, and the number of networks. We also make some **recommendations** based on the comparison.
>
> ```
> Comparing mini-batch-wise SS and epoch-wise SS.
> ```
>
> In Section 6.1, we demonstrated the benefit of mini-batch-wise SS vs. epoch-wise SS by comparing DivideMix and DivideMix+. We also compare GPL (mini-batch) with GPL (epoch) with a batch size of 256 on CIFAR-10. The results of test accuracy are as follows:
>
> | Method | Sym-40% | Sym-80% | Asym-40% |
> | :----| :----: | :----: | :----: |
> | GPL (epoch)| 93.13 | 88.96 | 91.55 |
> | GPL (mini-batch)| **94.09** | **91.83** | **93.06** |
>
> We observe that the test accuracy of GPL (mini-batch) constantly outperforms GPL (epoch) and have added the results in Appendix E.4. As for the detailed analysis of the pros and cons of our mini-batch-wise SS, please kindly refer to our reply to Q1 and Q3 in the general response.
>
>  - **Our recommendation**: If we choose **a bigger batch size**, our **mini-batch-wise SS** can be effective for both symmetric and asymmetric noise. From the sensitivity analysis on the batch size, the **batch size needs to be bigger than 64** to avoid the fluctuation of the noisy ratio in each mini-batch. Thus, in the following comparisons, we only use mini-batch-wise SS with a batch size of 256.
>
> ```
> Comparing  # of networks.
> ```
>
> We compare six instantiations of our framework on CIFAR-10: DivideMix (1 network), DivideMix (2 networks), DivideMix+ (1 network), DivideMix+ (2 networks), GPL (1 network), and  GPL (2 networks). "1/2 network/networks" means the instantiation without/with label co-refinement and co-guessing. The results of test accuracy and average training time are in the table below. We observe that training with two networks provides **a performance boost** for the corresponding instantiations. The performance at the **high noise rate is more significant than that of the medium noise rate**. This is reasonable since the purpose of two diverged networks is to avoid error accumulation [1] during the training, a high noise rate will induce more errors for the single model. We also made an apples-to-apples comparison between DivideMix+ (2 networks) and GPL (2 networks). The choice of SS and SSL will be discussed further in the following.
>
> | Method | Sym-40% | Sym-80% | Asym-40% | Average training time |
> | :----| :----: | :----: | :----: | :----: |
> | DivideMix (1 network)| 92.38 | 87.59 | 86.76 | 6.2 h |
> | DivideMix (2 networks)| **93.76** | **89.33** | **90.66** | 9.5 h |
> | DivideMix+ (1 network)| 92.50 | 90.66 | 89.36 | 5.4 h |
> | DivideMix+ (2 networks)| **93.82** | **92.04** | **91.37** | 8.8 h |
> | GPL (1 network)| 94.09 | 91.83 | 93.06 | 3.5 h |
> | GPL (2 networks)| **94.68** | **93.38** | **93.75** | 5.2 h |
>
>  - **Our recommendation**: Using two networks can **boost performance** for the corresponding instantiations, especially at high noise rates. The shortcoming of using two networks is the **increase in training time**. So it might not be necessary at low noise rates or on simple datasets (MNIST). In the following comparisons on CIFAR-10, we will use two networks for all instantiations for a fair comparison.

---

> > ### Author Response · Authors · 2022-06-26
> > **Response to Reviewer e8Ug (2/3)**
> >
> > ```
> > Comparing  different SS + SSL instantiations.
> > ```
> > The components considered in this work are SS = {GMM, SPD} and SSL = {Temporal Ensembling, MixMatch, Pseudo-Labeling}. We combine each SS and each SSL method to conduct a **systematic comparison**. We denote these instantiations as "SS" + "SSL". Please be noted that minibatch-wise SS and two networks are used in all the instantiations according to previous empirical results.
> >
> > The results of test accuracy on CIFAR-10 Sym 40% are as follows:
> >
> > | SS\SSL| Temporal Ensembling | MixMatch | Pseudo-Labeling |
> > | :----| :----: | :----: | :----: |
> > | GMM| 85.79 | 93.82 | 94.68 |
> > | SPD| 84.60 | 92.68 | 94.33 |
> >
> > The results of test accuracy on CIFAR-10 Sym 80% are as follows:
> >
> > | SS\SSL| Temporal Ensembling | MixMatch | Pseudo-Labeling |
> > | :----| :----: | :----: | :----: |
> > | GMM| 54.37 | 92.04 | 93.38|
> > | SPD| 51.56 | 90.98 | 92.20 |
> >
> > The results of test accuracy on CIFAR-10 Asym 40% are as follows:
> >
> > | SS\SSL| Temporal Ensembling | MixMatch | Pseudo-Labeling |
> > | :----| :----: | :----: | :----: |
> > | GMM| 64.53 | 91.37 | 93.75 |
> > | SPD| 68.74 | 92.11 | 92.14 |
> >
> >
> > From the above empirical results, we discovered some **general rules** and **synergistic pairs** when combining different SS and SSL methods using our framework:
> >
> > (i) General rules regarding SS: Under symmetric noise, instantiations with GMM achieve higher test accuracy than instantiations with SPD, while the results are reversed under asymmetric noise. This is because GMM distinguishes clean and noisy samples based on their loss distribution. For asymmetric noise, most samples have near-zero normalized loss due to the low entropy predictions of the network [1], causing performance deterioration for GMM. Since SPD leverages the prediction of its network to choose clean samples, its performance is more sensitive to noisy rates rather than noise type.
> >
> > - **Our recommendation**: For symmetric noise and high noise ratios, GMM should be used as the SS strategy. For asymmetric noise, SPD should be used as the SS strategy. For low noise ratios, SPD might be **more computationally efficient** (Table 7) in achieving competitive test accuracy.
> >
> > (ii) General rules regarding SSL: Instantiations with Temporal Ensembling are much worse than those with MixMatch and Pseudo-Labeling because Temporal Ensembling only uses consistency regularization for unsupervised loss. In general, instantiations with Pseudo-Labeling achieve higher accuracy than the other two because the regularization term used in Pseudo-Labeling prevents the model from assigning all labels to a single class at the early training stage. From the results of Asym 40%, we observe performance degrade of all SSL methods, especially Temporal Ensembling, combined with the same SS method (even though SPD can alleviate the harm of asymmetric noise). This is because the class distribution in the labeled/unlabeled sets after the SS process is class-imbalanced under the asymmetric noise. However, all the methods considered in this paper assume that the class distribution of labeled and/or unlabeled data is balanced.
> >
> > - **Our recommendation**: We would respectively argue that **our guidance is to select a suitable SSL method** for the corresponding labeled/unlabeled sets after the SS process instead of "using the strongest SSL algorithm" to achieve the best test accuracy. For example, if the class distribution of labeled/unlabeled sets is (close to) balanced, we can choose more advanced mainstream SSL methods [4, 5] as long as we take into account **computational resources and efficiency**. If labeled/unlabeled sets are class-imbalanced, we can choose SSL methods specifically designed to tackle class-imbalance data [6,7].
> >
> > (iii) Synergistic combination: We surprisingly find that **GMM + Pseudo-Labeling**, the two-network version of the original GPL, achieves the best performance in all settings, including the asymmetric noise **where SPD is supposed to be stronger**.
> >
> > - **Our recommendation**: GMM + Pseudo-Labeling achieves the best performance for both symmetric and asymmetric noise. We believe its superior performance benefits from the conceptually simple idea of Pseudo-Labeling to reduce confirmation bias generated from the SS process. Please be noted that Pseudo-Labeling was not published at top conferences. That being said, **we will keep exploring the possibility of more combinations of SS and SSL based on our framework to figure out the chemistry in between, instead of just simply piling up SOTA SS and SSL published in top conferences**.
> >
> > We have uploaded a new version of the paper that contains the above discussions in Appendix G.

---

> > > ### Author Response · Authors · 2022-06-26
> > > **Response to Reviewer e8Ug (3/3)**
> > >
> > > ### **C2** *Regarding the use of SS and SSL: if the clean examples are incorrectly identified then the network could inadvertently ignore labels that actually are meaningful, and vice versa, i.e. the network could learn from meaningless noise rather than clean labels. This could have catastrophic effects depending on the application*.
> > >
> > > Thank you for this valuable suggestion. We formalize a versatile framework that leverages SS to select clean samples and SSL to fully use the noisy samples by removing their labels. And we believe it will substantially impact **labor-intensive jobs of checking data label quality**, such as training models from the **web-crawled images** [2] and **medical data analysis** [3]. One potential risk caused by the wrong sample selection process is the increased chances of the model over-fitting **potential outliers** in the data that may lead to erroneous or misleading results, i.e., the misdiagnosing of patients. In the future, we aim to develop more precious and safer SS strategies to alleviate this negative impact. We have added the above discussion in Section 6.6.
> > >
> > > [1] DivideMix: Learning with Noisy Labels as Semi-supervised Learning, ICLR 2020
> > >
> > > [2] Harvesting image databases from the web, TPAMI 2010
> > >
> > > [3] Deep learning for healthcare: review, opportunities and challenges, Briefings in bioinformatics 2018
> > >
> > > [4] Fixmatch: Simplifying semi-supervised learning with consistency and confidence, NeurIPS 2020
> > >
> > > [5] FlexMatch: Boosting Semi-Supervised Learning with Curriculum Pseudo Labeling, NeurIPS 2021
> > >
> > > [6] CReST: A class-rebalancing self-training framework for imbalanced semi-supervised learning, CVPR 2021
> > >
> > > [7] Distribution aligning refinery of pseudo-label for imbalanced semi-supervised learning, NeurIPS 2020

---

### Author Response · Authors · 2022-06-26
**Response to All Reviewers (1/2)**

We would like to thank every reviewer for your time and constructive comments! We appreciate that reviewers acknowledge the simple intuition of our framework (Reviewer TbTQ, 8Gpa, e8Ug), the superior performance of our instantiations (Reviewer e8Ug), and the importance of the mini-batch SS (Reviewer e8Ug). The paper is well-written (Reviewer e8Ug) and provides thorough experiment results (Reviewer TbTQ, e8Ug). We also notice that there exist some common concerns on the reasons why the mini-batch SS should work compared to the batch-wise SS (**Q1**), more large-scale real-world datasets (**Q2**), and the discussion of limitations of the proposed approach (**Q3**), which we will address in the following:

### **Q1** (Reviewer TbTQ, 8Gpa) *The main difference between the proposed SS and related work is mini-batch sample selection. It is not clear why this method should work compared to the other baseline.*
The **advantages of mini-batch SS** over epoch-wise SS are:

- **Avoid confirmation bias**. In the case of epoch-wise SS, the divided clean/noisy sets are incorporated into the SSL phase and will not be updated till the next epoch. Thus, the confirmation bias induced from those wrongly divided samples will accumulate within the whole epoch. Our mini-batch SS strategy divides each mini-batch of samples into clean/noisy batches right before updating the network using SSL backbones. In the next mini-batch, the updated network can know better to distinguish clean and noisy samples, alleviating the confirmation bias mini-batch by mini-batch.

- **Improve computational efficiency**. Since the time complexity of most SS methods (including GMM and self-prediction divider) is not linear, the number of operations increases dramatically as the input size increases. Table 7 compares the training time of DividMix+ (mini-batch-wise) and DivideMix (epoch-wise) on CIFAR-10, showing DivideMix+ is more computationally efficient than DivideMix in both the SS process and the whole training process.

- **Inject stochasticity**. During the epoch-wise SS process, the model tends to select the confident samples that have been selected in previous epochs due to the model overfitting their labels. In this way, some confident but noisy samples will keep being selected by the model, resulting in performance degradation. The mini-batch-wise SS can inject stochasticity in training since each mini-batch of data is randomly sampled from the whole dataset, avoiding the model constantly selecting the same confident samples.

- **Up-to-date model for SS**. The mini-batch-wise usage of SS and SSL makes the data selection up-to-date. The model used to select the clean samples is updated using the SSL method at each minibatch. In comparison, the epoch-wise usage of the SS process selects the clean samples based on the model trained by SSL from the last epoch.

We have uploaded a new version of the paper that contains the advantages of mini-batch-wise SS in Section 3.1.


### **Q2** (Reviewer TbTQ, 8Gpa) *More real-world datasets should be compared.*
We test two more real-world datasets, Food-101N[1] and ANIMAL-10N[2]. Food-101N is a dataset for food classification. It consists of 310,009 training images and 25,000 testing images in 101 classes collected from the web. The estimated label purity is 80%. We use ResNet-50 pre-trained on ImageNet and there are 30 epochs in total. ANIMAL-10N contains 10 animals with confusing appearances downloaded online. There are 50,000 training and 5,000 testing images. The estimated label noise rate is 8%. We use VGG-19 with batch normalization and there are 100 epochs in total.

The results of Food-101N are as follows:

| Method | Test Accuracy |
| :----| :----: |
| Cross-Entropy| 81.53 |
| CleanNet [3]| 83.95 |
| DeepSelf [4]| 85.10 |
| DivideMix[5]| 85.64 |
| DivideMix+ (ours)| **86.93** |
| GPL (ours)| 86.59 |

The results of ANIMAL-10N are as follows:

| Method | Test Accuracy |
| :----| :----: |
| Cross-Entropy| 79.4 |
| Dropout [6]| 81.3 |
| SELFIE[2]| 81.8 |
| DivideMix | 82.4 |
| DivideMix+ (ours)| **83.6** |
| GPL (ours)| 83.2 |

As shown in the above tables, we can conclude that our instantiation, DivideMix+, consistently outperforms all base methods, including its epoch-wise variant, DivideMix. GPL achieves competitive results **using a single model**. We have uploaded a new version of the paper that contains more real-world datasets in Appendix E.1.

---

> ### Author Response · Authors · 2022-06-26
> **Response to All Reviewers (2/2)**
>
> ### **Q3** (Reviewer TbTQ, e8Ug) *No discussion of limitations of the proposed method.*
> ```
> Limitations Regarding mini-batch SS.
> ```
> Indeed, noise ratios in different mini-batches inevitably fluctuate since each mini-batch of data is randomly sampled from the whole dataset. When the overall noise level is high and the batch size is too small, the SS strategy might have trouble distinguishing clean and noisy samples in some **severely corrupted mini-batches**, which may deteriorate the overall performance. For example, in DivideMix+, we apply GMM to each mini-batch of data. The fluctuation of the noisy ratio in each mini-batch might result in inconsistent data selection criteria (clean threshold). Moreover, we conduct a **sensitivity analysis on the batch size of mini-batch SS** of our two instantiations, DivideMix+ and GPL, on CIFAR-10 Sym 80%.
>
> | Method\Batch size | 32 | 64 | 128 | 256 | 512 |
> | :----| :----: | :----: | :----: | :----: | :----: |
> | DivideMix+| 90.36 | 91.41 | 91.97 | **92.04** | 91.63 |
> | GPL | 88.46 | 90.16 | 91.81 | **91.83** | 91.03 |
>
> We observe that larger the batch size achieves better test accuracy since extremely high noise mini-batch can be avoided. Too large batch size might result in performance degradation because the model might be trapped in local optimum [12].
>
> We have uploaded a new version of the paper that contains the sensitivity analysis in Appendix E.2.
>
> ```
> Limitations regarding SSL.
> ```
> Under the symmetric noise, the labels are uniformly corrupted. If the SS strategy can distinguish clean/noisy samples well enough, the class distribution in the labeled/unlabeled sets is uniform. While under the asymmetric noise, the labels are flipped to similar classes, resulting in the class-imbalanced labeled/unlabeled data. However, **traditional SSL methods [7,8,9] (including the SSL backbones of our instantiations) assume that the class distribution of labeled and/or unlabeled data is balanced**. In Table 1 (symmetric noise), methods using SSL (DivideMix, DivideMix+, and GPL) outperform others in most cases. In Table 2(asymmetric noise), methods using SSL perform slightly worse than other SOTA methods in MNIST and FASHION MNIST. Please be noted that our framework can absorb the advantages of various SS and SSL methods. **To deal with the class-imbalanced issue in asymmetric noise, we can develop new instantiations using SSL methods designed to tackle class-imbalance data [10,11]**.
>
> We have uploaded a new version of the paper that contains the limitations of SS and SSL components in Appendix F.
>
> [1] Transfer learning for scalable image classifier training with label noise, CVPR 2018
>
> [2] Selfie: Refurbishing unclean samples for robust deep learning, ICML 2019
>
> [3] Cleannet: Transfer learning for scalable image classifier training with label noise, CVPR 2018
>
> [4] Deep Self-Learning From Noisy Labels, ICCV 2019
>
> [5] DivideMix: Learning with Noisy Labels as Semi-supervised Learning, ICLR 2020
>
> [6] Dropout: a simple way to prevent neural networks from overfitting, JMLR 2014
>
> [7] Pseudo-labeling and confirmation bias in deep semi-supervised learning, IJCNN 2020
>
> [8] Mix-match: A holistic approach to semi-supervised learning, NeurIPS 2019
>
> [9] Fixmatch: Simplifying semi-supervised learning with consistency and confidence, NeurIPS 2020
>
> [10] CReST: A class-rebalancing self-training framework for imbalanced semi-supervised learning, CVPR 2021
>
> [11] Distribution aligning refinery of pseudo-label for imbalanced semi-supervised learning, NeurIPS 2020
>
> [12] SGDR: Stochastic Gradient Descent with Warm Restarts, ICLR 2017

---

### Decision · Action_Editors · 2022-07-25

**Recommendation:** Accept with minor revision

**Comment:**

This manuscript proposes a semi-supervised learning framework with the presence of noisy labels, named SemiNLL. The proposed SemiNLL is sound, and the experimental results are shown sufficiently.

The AE has three minor suggestions: 1) Please proofread the entire paper to fix all the grammatical errors and typos; 2) the definition of noisy labels and the description of sample selection methods should be placed into the main text instead of the appendix; 3) make all the notations consistent throughout the entire paper.

Per the above comments, the AE therefore recommends “Accept with minor revision”.